# Learning the Hamiltonian of Disordered Materials with Equivariant Graph Networks

## Abstract

Graph neural networks (GNNs) have shown promise in learning the ground-state electronic properties of molecules and crystalline materials, subverting computationally intensive density functional theory (DFT) calculations. Materials with structural disorder, however, are more challenging to learn as they exhibit higher complexity and a more extensive palette of local atomic environments, all of which require large (10+ Å) cells to be accurately captured. In this work, we adapt efficient equivariant GNN approaches to learn disordered materials' electronic properties, represented by the Hamiltonian matrix ($\boldsymbol{H}$). Since creating a large graph corresponding to the whole structure of interest would be computationally prohibitive, we introduce an *augmented partitioning* approach in which the graph is sliced into multiple partitions, each augmented with masked virtual nodes and edges. This method maintains correct atomic neighborhoods within a single message passing layer, allowing for the network to learn the electronic properties of amorphous $HfO_2$ materials with 3,000 nodes (atoms), 500,000+ edges, and $\sim$28 million orbital interactions (non-zero entries of $\boldsymbol{H}$).

## 1 Introduction

Predicting structure-property relationships in atomically resolved materials lends itself optimally to graph-structured data. Graph Neural Networks (GNNs) have proven capable of learning these relationships while constrained by their underlying symmetries (Veličković et al. (2018)). Trained networks have been able to learn molecular- and atomic-level quantities (Wang et al. (2023)), circumventing otherwise computationally prohibitive simulations at the *ab initio* level. More recently, GNNs have been adapted to predict electronic properties, described by a discretized ground-state Hamiltonian matrix $\boldsymbol{H}$. Rapid and accurate constructions of $\boldsymbol{H}$ can unlock *in silico* explorations of the large design space of electronic materials (Klinkert et al. (2020)).

The matrix $\boldsymbol{H}$ can be decomposed into sub-matrices $\boldsymbol{H}_{i,j}$ that encode the coupling between the sets of atomic orbitals located on atoms $i$ and $j$. These coupling terms are a function of the identity and relative coordinates of the local environment. Predicting such electronically resolved information introduces additional challenges over atomically resolved quantities, as the output data is equivariant under rotation. Existing work on electronic property prediction has mainly treated the cases of small molecules (Zhong et al. (2023); Yu et al. (2023b); Bai et al. (2021)) and well-ordered materials (Li et al. (2022); Gong et al. (2023); Wang et al. (2024a)). The graph representations of these structures are fairly small - small molecules contain only a few atoms, and in crystalline materials all relevant structural information can be captured within the smallest repeating unit cell.

Many applications, however, require the computation of the electronic properties of materials with structural disorder, such as local or extended defects, or in an amorphous phase (Ducry et al. (2020); Kaniselvan et al. (2023); Strand et al. (2018)). These materials typically contain a limited number of atomic species, from 2 to 5, with quasi-random distributions. Accurate *ab initio* simulations of the resulting disordered atomic structures are only possible if large unit cells composed of hundreds to thousands of atoms are used (Repa & Fredin (2023)). As a consequence, prohibitively expensive computations must be performed with a density functional theory (DFT) tool to obtain their electronic properties. The prospect of applying deep-learning solutions to handle such materials is thus particularly attractive. To be of practical relevance, however, they should be able to generalize to large scales.

Here, we extend equivariant GNN approaches to learn and predict the electronic properties of materials in amorphous phases by fitting the sub-blocks of the ground-state Hamiltonian $\boldsymbol{H}$ in matrix form. Our main contributions are:

- We develop an efficient GNN-based model for electronic property prediction, by combining (1) the SO(2)-convolution approach detailed in Passaro & Zitnick (2023), (2) the equivariant attention mechanism introduced in Liao et al. (2023), and (3) concepts from Gong et al. (2023) and Wang et al. (2024a) to introduce learnable node/edge embeddings along with basis transformation layer to pre-process the targets and map predictions to the Hamiltonian output. We provide the code for this implementation in [the Supporting Materials].
- We propose an efficient augmented partitioning method that breaks down input graphs into small pieces and corrects atomic environments with masked virtual nodes and edges. This allows arbitrarily large graphs to be decomposed into independent partitions that can fit into GPU memory during training without compromising the achievable testing accuracy. Our approach enables the training and prediction of unfeasibly large systems including realistic amorphous materials and heterostructures that can contain up to hundreds of thousands of atoms in a unit cell.

We combine our model and *augmented partitioning* approach to treat a real example with practical scientific relevance. Specifically, we consider hafnium dioxide ($HfO_2$), one of the most technologically relevant amorphous oxides (Choi et al. (2011)). The theoretical study of defects and transport properties in $HfO_2$ is relevant to several research areas, from optimizing gate dielectrics for transistors (Strand et al. (2018)) to developing new resistive-switching technologies enabling in-memory computing (Kaniselvan et al. (2023)). With this, we achieve a prediction accuracy of $5.87\ meV$, matching the eigenvalues of $\boldsymbol{H}$ to within 0.87% relative L1 error, on structures with 3,000 atoms, which require several (3.65) hours to compute using DFT. Our work advances applications of equivariant GNNs towards practical use cases in computational physics, chemistry, and materials science.

## 2 BACKGROUND & RELATED WORK

The electronic properties of a material refer to its set of energy levels ($\varepsilon$) and wavefunctions ($\psi$) that electrons can occupy. They correspond to the eigenvalues and eigenvectors of the Hamiltonian matrix $\boldsymbol{H}$ describing the atomic system of interest. This quantity is a function of the location (relative positions $\{\boldsymbol{r}_i\}$) and identity (atomic numbers $\{Z_i\}$) of all constituent atoms $\{i\}$ (Hohenberg & Kohn (1964)). Therefore, predicting the electronic properties consists of learning the mapping $F : \{\boldsymbol{r}_i, Z_i\} \rightarrow \boldsymbol{H}$ between the atomic structure and the elements of the corresponding Hamiltonian matrix (**Fig. 1**).

The entries of the ground-state Hamiltonian matrix $\boldsymbol{H}$ are typically computed from first-principles with DFT (Kohn & Sham (1965)). In several widely used codes, the wavefunctions are expanded into a basis $|\varphi\rangle$ of nonorthogonal atomic orbitals localized around atomic positions, each built, for example, from contracted Gaussian functions (Kühne et al. (2020); Neese (2011)). These orbitals transform like spherical harmonics under rotation $\hat{\boldsymbol{r}} \rightarrow \hat{\boldsymbol{r}}'$: $Y_m^l(\hat{\boldsymbol{r}}') = \sum_{m'} \boldsymbol{D}_{mm'}^l(\boldsymbol{R}) Y_{m'}^l(\hat{\boldsymbol{r}})$. Here, $Y_m^l$ is the spherical harmonic of degree $l$ and order $m \in \{-l, \ldots, l\}$. $\boldsymbol{D}_{mm'}^l(\boldsymbol{R})$ is the Wigner-D matrix of degree $l$ corresponding to the rotation $\boldsymbol{R}$, which transforms the corresponding spherical harmonic. $\hat{\boldsymbol{r}}$ and $\hat{\boldsymbol{r}}'$ are normalized direction vectors.

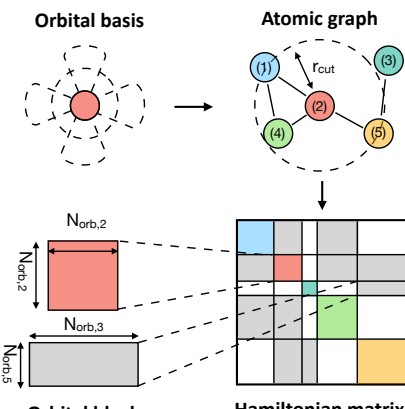

Figure 1: Schematic of the mapping between the atomic graph and the blocks of the Hamiltonian matrix $\boldsymbol{H}$ in the localized orbital basis of choice. Each orbital block represents the couplings between atomic orbitals on the same atom ($\boldsymbol{H}_{i,i}$, diagonal) or between two different atoms within $\mathrm{r}_{cut}$ ($\boldsymbol{H}_{i,j}$, off-diagonal).

The localized nature of the basis states leads to finite spatial overlaps between them. The resulting Schrödinger equation at the core of DFT takes the form of a generalized eigenvalue problem: $\boldsymbol{H}\psi = \varepsilon\boldsymbol{S}\psi$. Here, the Hamiltonian matrix $\boldsymbol{H}^{(N \times N)}$ has entries $\boldsymbol{H}_{i,j} = \langle\varphi_i|\hat{H}(\boldsymbol{r})|\varphi_j\rangle$ where $\hat{H}(\boldsymbol{r})$ is the so-called Hamiltonian operator, while the Overlap matrix $\boldsymbol{S}^{(N \times N)}$ is made of $\boldsymbol{S}_{i,j} = \langle\varphi_i|\varphi_j\rangle$. They

are both coarse-grained matrices of size $N = \sum_k N_{atoms}^k \cdot N_{orb}^k$, where $N_{atoms}$ is the number of atoms, $N_{orb}$ the number of orbitals per atom, and $k$ indexes over the different atomic species found in the system. Note that $\boldsymbol{S}$ reduces to the identity matrix in case of an orthogonal basis $|\varphi\rangle$. Otherwise, it can be directly computed from the basis as the problem's physics does not influence it.

The Hamiltonian matrix can be decomposed into sub-matrices $\boldsymbol{H}_{i,j}$ of size $(N_{orb}^i \times N_{orb}^j)$, each describing the interactions between all basis elements (orbitals) on atoms $i$ and $j$. Diagonal blocks $(\boldsymbol{H}_{i,i})$ are the interactions between orbitals on the same atom. When represented on a local basis, the matrix is near-sighted; the interactions between orbitals on different atoms decay exponentially with increasing interatomic distance. Since an atomic orbital basis is used, the sub-matrices are equivariant under rotation of the atomic bonds, with their transformation properties related by the Wigner-D matrix.

## 2.1 CHALLENGES UNIQUE TO DISORDERED MATERIALS

Computing the electronic properties of disordered materials with DFT still requires defining a repeating 'unit cell' and using periodic boundaries to avoid dangling bonds. This periodicity, however, can alter the material's amorphous nature if the repeating unit cell is too small. Atoms can interact with all their periodic images, leading to non-physical phenomena such as the formation of coherent electronic states across cells. These phenomena can be prevented by constructing 'large-enough' unit cells (12+ Å (Repa & Fredin (2023)) to a few nanometers (Ducry et al. (2020))) that better approximate disorder. Generating the Hamilto-

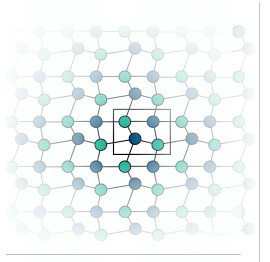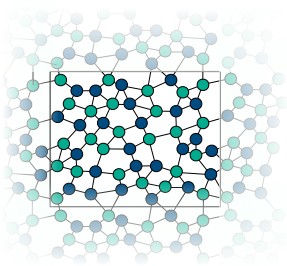

Figure 2: Illustration of the difference between ordered (left) and disordered materials (right), whose structural features can only be captured by defining a large unit cell. In both cases, the smallest repeating unit cell is delimited by a black box, while the circles/lines correspond to atoms/bonds.

nian matrix $\boldsymbol{H}$ of these systems with DFT involves tens to hundreds of self-consistent field (SCF) loops, each requiring the diagonalization of an intermediate $\boldsymbol{H}$. As this numerical operation scales with $\mathcal{O}(N_{atoms}^3)$, analyzing electronic properties for large amorphous systems (or different amorphous representations of the same material) is very often computationally unaffordable.

## 2.2 DEVELOPMENT OF MODELS FOR THE PREDICTION OF ELECTRONIC PROPERTIES

Only a few studies have attempted to directly predict the Hamiltonian matrix $\boldsymbol{H}$ rather than directly fitting invariant quantities such as the total energy. The key is to constrain the solution space by leveraging prior knowledge of physical symmetries, e.g., rotational equivariance of orbital blocks.

Several works leverage local structure descriptors containing sufficient detail about each atom's environment. The mapping between these descriptors and the orbital blocks of the Hamiltonian was then learned through Kernel Ridge Regression (Hegde & Bowen (2017)) or multilayer neural networks (Schütt et al. (2019); Gu et al. (2024)). Certain descriptors, such as Atomic Cluster Expansion (ACE) (Drautz (2020)), can be extended to orbitally resolved data (Nigam et al. (2022)) and used to predict the Hamiltonian matrix of crystalline materials (Zhang et al. (2022)). These approaches achieved prediction accuracy of a few $meV$ on small-molecule datasets. Rotational equivariance was enforced through the descriptors (Zhang et al. (2022)) or data augmentation (Schütt et al. (2019)). Initial GNN-based approaches, such as the network developed by Li et al. (2022), are intrinsically invariant to the translation and permutation of the inputs. Information about the rotational equivariance was incorporated by rotating to a pre-selected axis before training, which reduces the problem to a rotationally invariant one.

In equivariant GNNs, the predicted Hamiltonian rotates along with the input (Yu et al. (2023b); Zhang et al. (2024); Batatia et al. (2023); Gong et al. (2023)), which requires maintaining SO(3)-equivariance within the model. This means that all network operations $f$ acting on input embedding $\boldsymbol{x}^l$ of degree $l$ must satisfy: $f(\boldsymbol{D}^l(\boldsymbol{R}) \cdot \boldsymbol{x}^l) = \boldsymbol{D}^l(\boldsymbol{R}) \cdot f(\boldsymbol{x}^l)$. The networks are trained using Message Passing (MP), where each MP layer works as follows: An atom $i$ receives input messages

from each neighboring source atom $j$. Each input message goes through convolution operations that combine features with different $l$ while preserving equivariance; a specific output embedding $\boldsymbol{x}_{ji}^{l_3}$ of degree $l_3$ can be computed through: $\boldsymbol{x}_{ji}^{l_3} = \sum_{l_1, l_2} \boldsymbol{x}_j^{l_1} \otimes_{l_1, l_2}^{l_3} h_{l_1, l_2, l_3} Y^{l_2}(\hat{\boldsymbol{r}}_{ji})$. Here, $\hat{\boldsymbol{r}}_{ji}$ is a normalized vector indicating the direction of the edge connecting the atoms $j$ and $i$, and $h$ is a set of trainable weights. The sum runs over tensor products which take $\boldsymbol{x}_j$ (a source input embedding of degree $l_1$) and $Y^{l_2}$ (a filter spherical harmonic embedding of degree $l_2$) and produce the output embedding:

$$(\boldsymbol{x}_j^{l_1} \otimes_{l_1, l_2}^{l_3} Y^{l_2}(\hat{\boldsymbol{r}}_{ji}))_{m_3}^{l_3} = \sum_{m_1, m_2} (\boldsymbol{x}_j^{l_1})_{m_1} C_{(l_1, m_1), (l_2, m_2)}^{l_3, m_3} h_{l_1, l_2, l_3} Y_{m_2}^{l_2}(\hat{\boldsymbol{r}}_{ji}),$$

where the $C_{(l_1, m_1), (l_2, m_2)}^{l_3, m_3}$ are the Clebsch-Gordan coefficients that are indexed by the order $m$ and degree $l$ of the input, filter, and output embeddings. The combination of feature $(x)$ and geometric $(\hat{r})$ information along each edge encodes both the identity and structure of the system. These 'Tensor Field Networks' (TFNs) (Thomas et al. (2018)) achieve state-of-the-art accuracy on small molecule (Yu et al. (2023b)) and crystalline (Gong et al. (2023)) datasets. However, they are also much more computationally expensive. The network training scales with $\mathcal{O}(l_{max}^6)$, where $l_{max}$ is the maximum degree of the angular momentum considered. Fully, E(3)-equivariant networks are difficult to apply beyond a few atoms (Zhang et al. (2024)).

Recently, the computational cost of training equivariant GNNs has been significantly reduced by combining the benefits of data rotation and equivariant network operations. These approaches take advantage of the fact that when edges are rotated to align with a fixed axis ($y$ or $z$ depending on convention), the only non-zero spherical harmonic components are those of order $m = 0$. By keeping track of the bond vectors and performing internal spherical rotations, complex SO(3) convolutions can thus be reduced into SO(2) linear convolution operations (Passaro & Zitnick (2023)). Additionally, under these conditions, the Clebsch-Gordan coefficients exhibit a predictable sparsity pattern (non-zero only when $m_3 = \pm m_1$). Altogether, the scaling reduces to $\mathcal{O}(l_{max}^3)$ (Wang et al. (2024a)), and the network's training speeds up, enabling the use of higher-order angular momenta ($l_{max}$) and more parameters to capture finer, more complex details of the surrounding environment. An advanced SO(2) convolution network was developed by Passaro & Zitnick (2023) (eSCN)) and further expanded by Liao et al. (2023) (EquiformerV2) with the inclusion of equivariant attention and separable activation layers. A subsequent implementation of this approach on Hamiltonians by Wang et al. (2024a) achieved better performance on custom crystalline 2D-material datasets compared to previous tensor field and invariant networks.

## 3 METHODS

We adapt the 'EquiformerV2' network by incorporating concepts from Gong et al. (2023) and Wang et al. (2024a). In this section, we present an overview of the methods used to initialize the graph, construct the network, and propose an efficient *augmented partitioning* approach to train it. Relevant implementation details and ablation studies are presented in **Section 4** and **Appendices A, C, D**.

### 3.1 NETWORK LAYOUT

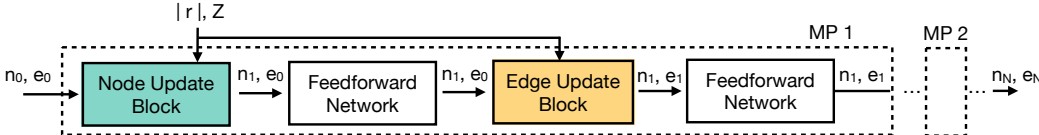

Figure 3: High-level overview of the network, illustrating the update of node features ($\boldsymbol{n}_X$) and edge features ($\boldsymbol{e}_X$) after message passing layer X (MP X). Z refers to atomic numbers, while $r_{ij}$ is the set of scalar distances between atoms.

When constructing a given material's graph, we can leverage the Hamiltonian's near-sightedness to only retain edges corresponding to atoms within a fixed interaction distance $r_{cut}$, beyond which the interactions are negligible. Orbitals located on different atoms can interact with each other over distances of $\sim$10 Å, giving rise to specific off-diagonal blocks in the Hamiltonian matrix.

The graph's nodes/edges are initialized with an embedding of shape $(N_n, (l_{max} + 1)^2, d_{sphere,n})/(N_e, (l_{max} + 1)^2, d_{sphere,e})$, where $N_{n/e}$ is the number of nodes/edges, $d_{sphere,n/e}$ the channel dimension for embeddings, and $l_{max}$ the maximum degree of the features. The $l = 0$ channels of the node embeddings are initialized with atomic numbers, while the $l = 0$ of the edge embeddings are initialized with the scalar distance between the two connecting nodes, expanded in the chosen basis, here Gaussian functions. All other components are initially set to 0. The orbital blocks representing the interaction between atoms are flattened into 1D tensors to form the labels for each node/edge during the supervised training process. They are then converted from uncoupled to coupled basis using a Wigner-Eckart transformation (**Appendix A.1**).

During training, each MP layer updates the node, and then the edge representations. Within the node update block, each node $i$ receives messages from each of its neighbors $j$, consisting of the concatenated embeddings $\boldsymbol{n}_i$, $\boldsymbol{n}_j$, and that of the edge $\boldsymbol{e}_{ji}$, rotated to align with the $z$-axis. After SO(2) convolutions are performed on the input messages, the resulting output messages are rotated back to their original orientations and aggregated onto the node $i$ to update its embedding $\boldsymbol{n}_i$. The updated node embeddings are then used to update the edges through a similar process, without the attention layer. Subsequently, the node embeddings in the next message-passing layer are updated with the new edge embeddings. A high-level overview of the network is presented in **Fig. 3**, and a more detailed outline can be found in **Appendix A**. During the inference phase, the output is converted back to the uncoupled basis through a Wigner-Eckart layer (**Appendix A.1**) to reconstruct the Hamiltonian matrix **H**.

Our network architecture is similar to that of EquiformerV2 and DeepH2. A particular difference lies in the use of gate activation layers instead of the $S^2$ activation layer of EquiformerV2. Although the $S^2$ layer allows for non-linearity to be introduced to higher-order features, achieving numerically perfect equivariance requires a large grid size, making it computationally expensive when implemented on large graphs (Passaro & Zitnick (2023)).

### 3.2 AUGMENTED PARTITIONING WITH MASKED VIRTUAL NODES/EDGES

Training GNN representations of the large amorphous materials considered here incurs high memory consumption and long computation times per epoch (**Appendix G.2**). The dense connectivity of these graphs also leads to heavy communication overhead in full-batch distributed approaches (Wan et al. (2022)).

A variety of methods have been developed to reduce the memory requirements and increase the amount of parallelism during training of large graphs (Besta & Hoefler (2024)). However, they do not cover the specificity of our application. Much of previous work is concerned with modifying connectivity to effectively propagate long-range information, for example, by dividing the graph into sub-graphs and passing messages intra- and inter-subgraph (Liao et al. (2018)), or introducing virtual nodes to transfer messages over longer distances (Qian et al. (2024)). Such approaches are unnecessary in our case since the graph is nearsighted and long-range interactions are negligible (**Appendix F.1**).

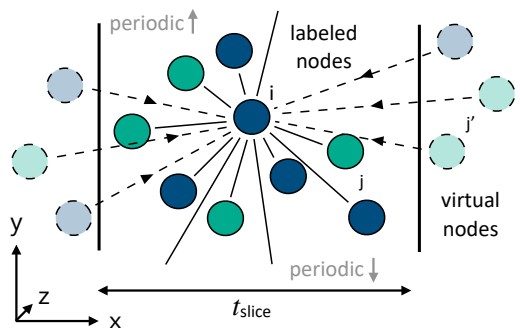

Figure 4: Illustration of how connectivity beyond the partition boundaries is incorporated, using one-way masked virtual edges ($e_{j' \rightarrow i}$) from masked virtual nodes ($j'$) to the labeled nodes. Edges between nodes within the partition are two-way ($e_{j \leftrightarrow i}$). A set of edges from labeled and virtual nodes is shown for a single node ($i$) within the partition. The solid vertical lines are the partition boundaries. Different colors represent different atomic species.

Neighborhood sampling techniques such as ClusterGCN (Chiang et al. (2019)) represent an alternative, but they sacrifice the exact connectivity of the graph to enable computational efficiency and scalability. As $H$ is a function of the relative positions and species of all constituent atoms, directly changing the representation of local atomic neighborhoods compromises the achievable accuracy. Omitting any connections leads to misinformation and poor generalization, as the network tries to fit to the target data while aggregating information through an incorrect/incomplete graph structure.

Training of large GNNs for electronic property prediction thus necessitates new strategies to become computationally feasible.

To enable efficient training of the graph while maintaining correct atomic environments and neighboring edge connections, we introduce an *augmented partitioning* approach. A visual representation of it is provided in **Fig. 4**. The graph is partitioned into slices along the $x$-axis (longest dimension), and maintains periodic edges across the $y$- and $z$-boundaries. Each slice contains atoms and edges within a fixed interval between $x_0$ and $x_0 + t_{slice}$, where $x_0$ and $t_{slice}$ are the starting $x$-coordinate and length of the slice, respectively. Atoms outside of a given slice but present in the connectivity lists of those within are represented by virtual nodes (**Fig. 4** - dashed circles). They are connected to the slice using one-way virtual edges (**Fig. 4** - dashed lines). Details about the construction of partitions are given in **Appendix C**.

These virtual nodes/edges are initialized similarly to their labeled counterparts with input atomic numbers and distances. However, their outputs are masked and omitted from the loss computation during training, validation, and inference. Our masking approach differs from the one used in transductive learning schemes, where the objective is to learn the outputs of masked nodes/edges (Kipf & Welling (2016)). Here, we do not attempt to learn or predict their outputs. The purpose of masked nodes and edges here is to inform each partition of its full connectivity and thus provide a much closer representation of the graph topology. As the set of virtual connections used to augment each graph corresponds only to the 1-hop neighborhood, we include only 1 MP layer in the network. During message passing, the network can then learn an accurate and generalizable aggregation function when fitting to the output values of the labeled nodes and edges. Hence, the network trained on a batch of such slices can predict the full graph of an unseen test structure.

### 3.3 A-HfO$_2$ STRUCTURE CREATION

To generate sufficiently rich training data, existing datasets typically sample molecules at various time steps of molecular dynamics (MD) trajectories (Yu et al. (2023a); Schütt et al. (2019); Christensen & lilienfeld (2020)) or generate multiple small perturbations of the atoms in a crystalline lattice (Li et al. (2022)). In the case of amorphous crystals, we take advantage of the fact that (1) almost each node has a different local atomic environment, and (2) the structure contains a large sampling of different motifs. A wide range of training data can thus be captured within a single sample. We, therefore, generated a dataset of only three unique amorphous HfO$_2$ (a-HfO$_2$) structures for training, validation, and testing, respectively. The DFT-based Hamiltonian of each of these systems ($\boldsymbol{H}^{GT}$) was computed with a single-$\zeta$ valence (SZV) basis with 10 (4) Gaussian orbitals per Hf (O) atom. Details of the structure generation can be found in **Appendix F.1**. A structure example is shown in **Fig. 5**.

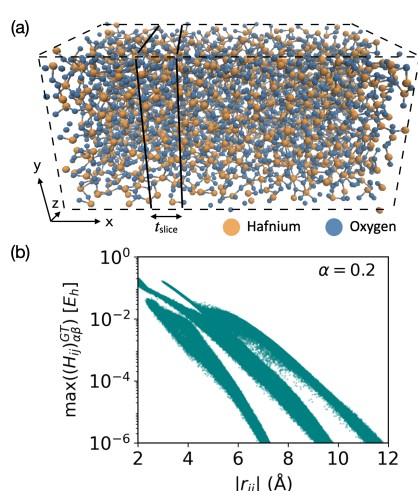

Figure 5: (a) Example atomic structure of a-HfO$_2$, showing oxygen atoms in blue and hafnium atoms in orange. The unit cell, which is illustrated by the black dashed box, has $x$-, $y$-, and $z$-dimensions of 53.876 Å $\times$ 26.308 Å $\times$ 26.242 Å, respectively. A slice (partition), characterized by a length $t_{slice}$, is also illustrated. (b) Distribution of the maximum element of each block of $\boldsymbol{H}$ as a function of interatomic distance, demonstrating the near-sighted nature of orbital interactions.

| Structure | # atoms | # orbitals | # edges | $x$ [Å] | $y$ [Å] | $z$ [Å] | $nnz_H$ |
|---|---|---|---|---|---|---|---|
| 1-validation | 3,000 | 18,000 | 527,348 | 52.876 | 26.308 | 26.242 | 28,625,310 |
| 2-training | 3,000 | 18,000 | 533,364 | 52.346 | 26.237 | 26.293 | 28,943,862 |
| 3-testing | 3,000 | 18,000 | 530,920 | 52.722 | 26.267 | 26.191 | 28,805,422 |

Table 1: Attributes of the three generated a-HfO$_2$ structures: The $[x, y, z]$ triplet defines the periodic unit cell size. $nnz_H$ is the number of non-zero elements in the Hamiltonian, encompassing all orbital interactions. Edges were defined according to an interaction distance of $r_{cut} = 8$ Å.

## 4 RESULTS

For fair comparisons in experiments where the quantity of training data varies, we used a ReduceLRonPlateau scheduler that tracks the validation loss per epoch and reduces the learning rate by a fixed decay factor when no further decrease is detected. Training is stopped once a minimum learning rate is reached. Details on the values of the hyperparameters and the scheduler settings for different experiments can be found in **Appendix E**.

When trained on the $(H_{i,j})_{\alpha,\beta}$ elements of $H_2O$ molecules from the MD17 dataset (Schütt et al. (2019)), our network as detailed in **Appendix A** can achieve prediction accuracy within an order of magnitude ($100 \times 10^{-6} E_h$ ($E_h$ = Hartree) vs. $\sim 10 \times 10^{-6} E_h$) of state-of-the-start equivariant GNN approaches, while using fewer layers (2 vs. 4-5) (**Appendix B**).

The treatment of the a-$HfO_2$ structures is, however, more challenging. We test the model's ability to generalize to different configurations of this system by predicting $H$ of the third structure in **Table 1**, which remains unseen during the training process. The *augmented partitioning* scheme is only applied during training (on structure 2), while the $H$ of the unseen structure as a single graph, including all nodes and edges, is predicted during inference (structure 3). Errors are reported separately for nodes ($\epsilon_n$) and edges ($\epsilon_e$) for a more complete analysis that distinguishes intra- and inter-atomic orbital interactions, which have very different magnitudes. Considering the large number of different motifs in the disordered structure, we also report the standard deviation of the node and edge errors ($\sigma_n$ and $\sigma_e$) to provide information on the consistency of the predictions.

### 4.1 CUTOFF RADIUS AND CONNECTIVITY

We first explore the minimal graph connectivity that can be used by the network to accurately learn relevant features. To do this we use the slice partition approach introduced in **Section 3.2**, using a single slice of length $t_{slice} = 3$ Å to train the network. Reducing the value of $r_{cut}$ below 8 Å noticeably increases the error ($\epsilon_{n/e}$), thus demonstrating the sensitivity of $H$ to the exact connectivity of the graph. Going from $r_{cut} = 8$ Å to 10 Å , the prediction error begins to plateau, but the node degree (which is proportional to the memory consumption of the network) grows by $1.7\times$. An $r_{cut}$ of 8 Å also results in negligible changes to the eigenvalue spectra (**Fig. 10** in **Appendix F.1**). We thus set $r_{cut}=8$ Å when defining graph edges for subsequent experiments. Note that $\epsilon_{node} >> \epsilon_{edge}$ as the magnitude of the node labels is $\sim 100\times$ larger than that of the edge labels.

| $r_{cut}$ [Å] | $deg(n)$ | $deg(n)'$ | Epochs | $\epsilon_n$ | $\sigma_n$ | $\epsilon_e$ | $\sigma_e$ |
|---|---|---|---|---|---|---|---|
| 4 | 20.99 | 10.81 | 13,816 | 4.14 | 0.00960 | 9.60 | 0.00105 |
| 6 | 74.23 | 27.78 | 14,071 | 3.79 | 0.00925 | 0.40 | 0.00074 |
| 8 | 177.09 | 51.17 | 18,245 | 3.76 | 0.00903 | 0.22 | 0.00050 |
| 10 | 346.03 | 81.25 | 22,463 | 3.76 | 0.00886 | 0.15 | 0.00040 |

Table 2: Prediction accuracy of the network with different $r_{cut}$. Training was done with a single slice of length $t_{slice} = 3$ Å taken from structure 2 at $x_0 = 25$ Å. The edge connectivity of the matrix is set by $r_{cut}$. $deg(n)$ is the average node degree, and $deg(n)'$ the reduced node degree omitting virtual node neighbors. Note that for this value of $t_{slice}$, the majority of neighbors for the average node are virtual. $\epsilon_n$ and $\epsilon_e$ are the Mean Average Error (MAE) for nodes/edges, respectively, and $\sigma_n$ and $\sigma_e$ are the corresponding standard deviations. All units are in [$\times 10^{-3} E_h$]. In all cases, one MP layer is used. The validation loss of the model is computed from a slice of similar length, interaction distance, and starting location extracted from structure 1. The networks are tested on an unseen full graph (structure 3) constructed with the same $r_{cut}$.

### 4.2 ABLATION STUDIES OF THE TRAINING APPROACH

Next, we perform a study on the design features of the *augmented partitioning* approach introduced in **Section 3.2**. In particular, we examine the influence of virtual nodes, one-way/two-way, and one/two MP layers on the prediction accuracy. **Table 3** compiles ablation studies of the three aforementioned parameters, using 18 slices of length $t_{slice} = \sim 3$ Å that cover the full training structure (structure 2).

Compared to training with raw partitions, the addition of virtual nodes and edges reduces both $\epsilon_{node}$ and $\epsilon_{edge}$ by over $\sim 50\%$ when evaluated on the full test structure. Such an improvement is

| Edges | # MP | $\epsilon_\mathbf{n}$ | $\sigma_\mathbf{n}$ | $\epsilon_\mathbf{e}$ | $\sigma_\mathbf{e}$ |
|---|---|---|---|---|---|
| $n' \quad n$ | 1 | 5.18 | 7.93 | 1.66 | 0.0026 |
| $n' \to n$ | 1 | 2.29 | 5.09 | 0.20 | 0.0038 |
| $n' \leftrightarrow n$ | 1 | 2.38 | 5.26 | 0.23 | 0.0042 |
| $n' \to n$ | 2 | 8.60 | 21.01 | 0.24 | 0.0043 |
| $n' \leftrightarrow n$ | 2 | 8.44 | 20.65 | 0.24 | 0.0041 |

Table 3: Ablation studies exploring the impact of virtual nodes, one-way edges, and # of MP layers on the prediction accuracy when tested on structure 3, using slices of length $t_{slice}$ = 3 Å from structure 2 for training and of length $t_{slice}$ = 4 Å (taken at $x_0$ = 25 Å ) from structure 1 for validation. The first column indicates the edge direction between virtual ($n'$) and labeled ($n$) nodes. $n' \to n$ are one-way (incoming) edges, while $n' \leftrightarrow n$ are two-way edges. Values are reported in $\epsilon_n[mE_h]$ and $\sigma_n[\mu E_h]$.

| $t_{slice}$ [Å] | $N_t$ | $N_e$ | Epochs | $\epsilon_\mathbf{n}$ | $\sigma_\mathbf{n}$ | $\epsilon_\mathbf{e}$ | $\sigma_\mathbf{e}$ |
|---|---|---|---|---|---|---|---|
| ~1 | 54 | 47,958 | 15,744 | 2.30 | 5.37 | 0.20 | 0.37 |
| ~2 | 27 | 95,398 | 15,628 | 2.32 | 5.19 | 0.20 | 0.36 |
| ~3 | 18 | 141,512 | 15,675 | 2.29 | 5.09 | 0.20 | 0.38 |
| ~4 | 14 | 184,730 | 14,833 | 2.45 | 5.47 | 0.21 | 0.38 |
| ~8 | 7 | 320,324 | 20,599 | 2.45 | 5.41 | 0.17 | 0.34 |
| ~12 | 5 | 381,504 | 19,351 | 2.59 | 5.86 | 0.18 | 0.36 |
| ~52 | 1 | 533,364 | 23,396 | 2.46 | 5.39 | 0.16 | 0.33 |

Table 4: Prediction accuracy when the network is trained on differently-sized partitions of the same graph (structure 2), using 1 MP layer. $N_e$ = # of slices, $N_e$ = total # of labeled edges. The total number of labeled nodes remains constant. The number of slices is equal to $\sim L/t$, where $L$ = 52.346 Å is the full length of structure 2 used for training. The validation set is a slice of length $t_{slice}$ = 4 Å starting at $x_0$ = 25 Å from structure 1. The models are tested on the full unseen structure (structure 3) and values are reported in $\epsilon_n[mE_h]$ and $\sigma_n[\mu E_h]$.

expected, as raw partitions are characterized by a large proportion of missing edges. We note that the best performing case achieves an $\epsilon_n$ and $\epsilon_e$ of 2.29 $\times 10^{-3} E_h$ and 0.20 $\times 10^{-3} E_h$, respectively, using a single MP layer and one-way edges from virtual to labeled nodes ($n' \to n$). As $t_{slice} < r_{cut}$, a one-hop neighborhood is sufficient to cover all nodes across a partition. This property renders a single MP layer sufficient, considering the near-sightedness of the Hamiltonian. The single MP-network's performance is trivially unaffected by the use of one-way virtual edges rather than two ($n' \leftrightarrow n$); virtual nodes do not need to aggregate information as they are omitted from the loss. The edges connecting the labeled to virtual nodes ($n \to n'$) can thus be omitted to reduce memory consumption. Using 2 MP layers with one layer of virtual nodes, however, degrades the performance as the 2-hop neighborhood is not correctly represented. It should be emphasized that the full graph with 2 MP layers does not fit into the memory of a single NVIDIA A100 GPU (requiring > 80 GiB). Results with 2 MP layers are in **Appendix D** (under **Table 9** ).

### 4.2.1 EFFECT OF AUGMENTED PARTITIONING ON PREDICTION ACCURACY

The *augmented partitioning* approach introduced in **Section 3.2** allows for the subdivision of the large graphs associated with the training of disordered materials by defining small slices (i.e., small $t_{slice}$) that can be treated sequentially on a single GPU or distributed across multiple GPUs and processed independently in parallel. Intermediate quantities do not need to be communicated during the forward/backward passes.

To establish that the network trained on augmented partitions does not suffer from loss of accuracy, we partition the same full graph into different numbers of slices with different thicknesses for training. In each case, the total number of labeled atoms summed up across all the slices remains the same (3,000), while the total number of labeled edges reduces with increasing partitions. From **Table 4** it can be seen that the prediction error is insensitive to partition size. Despite the different divisions ranging from 5 ($t_{slice}$ = ~12 Å) to 54 ($t_{slice}$ = ~1 Å) slices, $\epsilon_\mathbf{n}$ and $\epsilon_\mathbf{e}$ remain very close to the values obtained by training with the full graph ($t_{slice}$ = 52.346 Å). For small slices, the reduced fraction of labeled connections along the $x$ direction does not affect the accuracy as the remaining data along the $y$ and $z$ directions is sufficient to train the network. The combined MAE loss from all 3,000 atoms and 530,920 edges for the best performing case ($t_{slice}$ = ~3 Å) is 0.2159 $mE_h$, or 5.87 $meV$. This value is comparable to what a previous study obtained (2.2 $meV$) using equivariant GNNs for much smaller structures with ≤150 atoms per unit cell (Wang et al. (2024b)).

### 4.3 PERFORMANCE ON A-HfO$_2$

To assess whether the prediction accuracy of the trained network is sufficient for practical application, we assemble the full Hamiltonian of the a-HfO$_2$ test structure using the network outputs ($\boldsymbol{H}^{pred}$), extract key quantities, and compare them with results obtained from the ground-truth

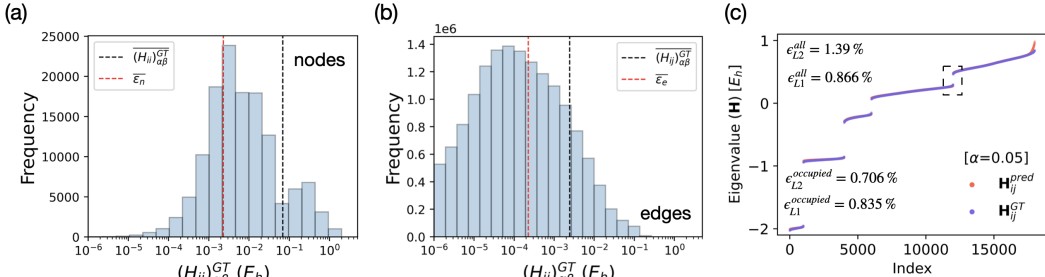

Figure 6: (a) $\epsilon_n$ of the predicted node blocks (red dashed lines) plotted against the full distribution of entries in the ground-truth Hamiltonian matrix $(H_{i,j})_{\alpha,\beta}^{GT}$. (b) Same as (a), but for $\epsilon_e$ of the predicted edge blocks. (c) Eigenvalue spectra of the predicted ($\boldsymbol{H}^{pred}$) and reference ($\boldsymbol{H}^{GT}$) Hamiltonian matrices. The alpha value indicates the scatter point transparency, such that the differences between the two scatter point sets can be more clearly seen. $(\boldsymbol{H}_{i,j})^{pred}$ is symmetrized before diagonalization with $\boldsymbol{H} = \frac{1}{2}(\boldsymbol{H} + \boldsymbol{H}^{\dagger})$. The relative L1/L2 errors in the eigenvalue spectra, computed as $(\|(\vec{E})^{pred} - (\vec{E})^{GT}\|_{norm}^2)/(\|(\vec{E})^{GT}\|_{norm}^2)$ (norm = 1, 2) where $\vec{E}$ is the vector of eigenvalues, are also shown for all eigenvalues and the ones corresponding to occupied states (below $0.305\ E_h$). The black dashed box indicates the bandgap, which is defined as the difference between the first energy above and below the Fermi level.

Hamiltonian ($\boldsymbol{H}^{GT}$) computed with DFT. In **Fig. 6** we show the achieved $\epsilon_n$ and $\epsilon_e$ relative to the magnitude of the $\boldsymbol{H}^{GT}$ entries. We then compute the eigenvalue spectrum of $\boldsymbol{H}^{pred}$ and $\boldsymbol{H}^{GT}$ as well as the error distribution between them. The network used to create $\boldsymbol{H}^{pred}$ was trained with 18 partitions of length $t_{slice} = \sim 3$ Å. We observe that both $\epsilon_n$ and $\epsilon_e$ are at least one order of magnitude lower than the average matrix element corresponding to the diagonal/off-diagonal blocks of the Hamiltonian (black dashed line). This allows for the reconstructed $\boldsymbol{H}^{pred}$ to reproduce all eigenvalues of $\boldsymbol{H}^{GT}$ within 0.87% relative L1 error. The error is 0.84% when eigenvalues of unoccupied states above the cutoff $0.306\ E_h$ are excluded. The remaining error is carried mostly by the largest eigenvalues, and distributed around the edges of energy gaps (see **Fig. 13** in **Appendix H**), which correspond to regimes of stronger inter-atomic orbital coupling (Atkins & De Paula (2009)).

## 4.4 OTHER EXAMPLES

To further demonstrate the strength and robustness of the *augmented partitioning approach* , we have conducted additional experiments on other datasets (found in Appendix I). There we showed that the approach generalizes well to HfO$_x$ structures with differently distributed vacancies, and the model achieves an even better prediction accuracy of 1.43 $meV$ for amorphous PtGe, compared to the 5.87 $meV$ achieved for HfO$_2$.

## 4.5 COMPUTATIONAL COST

Compared to a naive full-batch training of the graph, our method using just 8 augmented slices results in a 6.5× speedup per epoch (0.38 vs. 2.5 s), and a 7.2× decrease in memory consumption per rank (8.59 vs. 61.68 GiB). A more complete analysis is provided in **Appendix G.2**. This scaling behavior is limited only by the overhead introduced by the virtual nodes/edges, and the small computational load imbalance from partitioning. Further computational improvements could be achieved by combining the augmentation approach with optimized graph partitioning algorithms (Karypis & Kumar (1998)) while leveraging periodicity.

The extension of GNN-based predictions to large material systems could potentially save tremendous amounts of computational time. While DFT calculations to obtain the $\boldsymbol{H}^{GT}$ of small molecules (e.g., H$_2$O) take only a few seconds, the same operation for a-HfO$_2$ structures made of 3,000 atoms is computationally $\sim$100× heavier ($\sim$0.04 vs. $\sim$3.65 node hours, see **Appendix G**). More importantly, the GNN prediction unlocks the ability to consider much larger structures than the ones considered here, the inference phase scaling with $\mathcal{O}(N_{atoms})$ while DFT calculations are limited to $\mathcal{O}(N_{atoms}^3)$. The model could also serve as an initial guess to DFT packages to reduce the number of self-consistent field iterations that are required to obtain converged electron densities (Unke et al. (2021)).

## 5 CONCLUSION

We adapted equivariant GNNs to learn the electronic properties of amorphous materials, and introduced an *augmented partitioning* approach to break down and train the large graphs encountered when dealing with realistic structural disorder, without sacrificing accuracy. More generally, we proposed a method to tackle the training of atomic systems that require large, highly connected, and near-sighted graphs where a strictly local atomic environment is sufficient. The key is the addition of virtual nodes and edges connected to relatively small partitions that mimic their neighborhood. The method, demonstrated here on a-HfO$_2$, can be straightforwardly applied to other disordered materials, or adapted to learn their other rotationally-equivariant attributes such as vibrational properties, e.g., phonon dispersions (Fang et al. (2024)).

The resulting networks capture relevant properties of a-HfO$_2$ in sufficient detail to achieve few-$meV$ accuracy and reproduce features of practical relevance, such as the energy eigenvalues. However, the sub-millielectronvolt range is not currently reached, contrary to what has been demonstrated with small-molecule datasets (Yu et al. (2023b); Unke et al. (2021)). This shortcoming can most likely be attributed to a combination of greater dataset complexity in amorphous compounds, coarser resolution of the training data ($\epsilon_{SCF} = 1 \times 10^{-6} E_h$), and limited network size. Note that the *augmented partitioning approach* is also a general method that can also be adapted for use in other, more expressive network architectures, since it is mainly applied during graph construction. Further data generation, parameter optimization, and enabling of networks with increased expressiveness will be the next steps.

### 5.1 OUTLOOK & APPLICATIONS

The ability to learn the electronic properties of complex disordered materials unlocks notable applications in computational physics, chemistry, and materials science. Several compounds are used in their amorphous phase, as they often exhibit different properties from their crystalline equivalents. For example, a-SiO$_2$ as dielectric layer has been a key enabler of the metal-oxide-semiconductor technology (Nekrashevich & Gritsenko (2014)), IGZO, thanks to its large electron mobility, serves as channel of flexible transistors (Kamiya et al. (2010)), a-HfO$_2$ allows for the (non-)volatile storage of information, when placed between two metallic electrodes (Chan et al. (2008)), and GST can be used to write/store data optically (Pirovano et al. (2004); Kolobov et al. (2004)). Downscaling of materials also reveals structural effects similar to disorder, e.g., defects (Wilhelmer et al. (2022)), strain Parton & Verheyen (2006), or grain boundaries (Weitz et al. (2009)). All of them require large unit cells to be accurately described (Lany & Zunger (2008); Zhao et al. (2020)). Computationally expensive DFT calculations represent a bottleneck towards investigating the electronic properties of such systems in simulations. Recent advances in graph neural networks, combined with domain-specific innovations to train them, will enable such explorations.

### AUTHOR CONTRIBUTIONS

[anonymized]

### ACKNOWLEDGMENTS

[anonymized]

### REPRODUCIBILITY

The code used to set up and train the network is available at the following repository: [anonymized code temporarily provided as supplementary material]. CP2K was used to generate the custom HfO$_2$ dataset. Input files are provided in the same repository.

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

## A NETWORK ARCHITECTURE

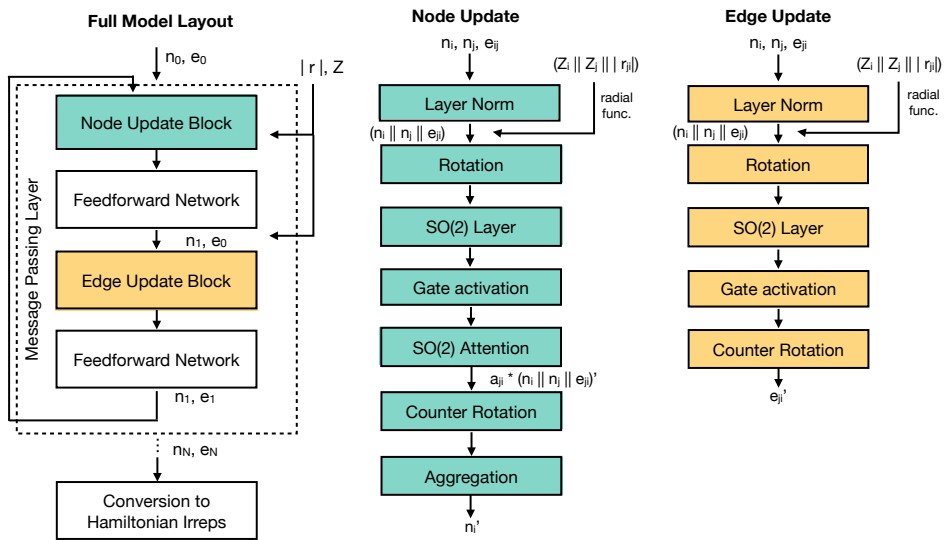

Figure 7: The node embeddings after the message passing layer labeled X are denoted as $\boldsymbol{n}_X$, and the corresponding edge features are $\boldsymbol{e}_X$. $\mathrm{a}_{ij}$ is the block of attention weights, $Z_i$ is the atomic number of atom $i$, and $\mathrm{r}_{ij}$ is the scalar distance between atoms $i$ and $j$. The input message to the update blocks consist of the concatenated embeddings of source and target nodes $j$ and $i$, along with their connecting edge $e_{ji}$. They are multiplied by a set of weights generated from the radial function using a scalar embedding that consists of the atomic numbers $Z_i$ and $Z_j$ concatenated with the edge distance $|r_{ji}|$

In this section, we provide further details on the architecture adapted from EquiformerV2 (Liao et al. (2023)) and DeepH2 (Wang et al. (2024a)), in addition to the network initialization description in **Section 3.1**.

During message passing, input messages in the form of node embeddings of size $(N_n, (l_{max}+1)^2, d_{sphere,n})$ and edge embeddings of size $(N_e, (l_{max}+1)^2, d_{sphere,n})$ from surrounding neighboring source nodes are passed into the target node. Note that for our network, $d_{sphere,n}$ and $d_{sphere,e}$ are set to the same value $d_{sphere}$, known as the number of spherical channels.

Inside the node update block, the source and target atom embeddings are concatenated together with the edge embeddings of their connecting edge to form an input message of size $(N_e, (l_{max}+1)^2, 3d_{sphere})$. The message is multiplied by weights generated by the radial functions using scalar embeddings (atomic numbers and distances), making it more receptive to small changes in environments. The dimension of these scalar embeddings are similarly set to $d_{sphere}$.

Afterwards, the input message is rotated to align with the z axis through the rotation block. The rotation block contains rotation matrices that were pre-computed using the normalized edge vectors of every edge in the graph. The rotated message is reshaped into $m$ major order for linear convolutions to be performed for each $m = 0$ to $m = m_{max}$ with the number of $l$ components for each m is given by $l_{max} - m + 1$. $m_{max}$ and $l_{max}$ ($m_{max} \leq l_{max}$) are other hyperparameters that can be adjusted. The convolutions produce an output embedding of size $(n_e, (l_{max}+1)^2, d_{attn\_hidden})$, where attention hidden channels $d_{attn\_hidden}$ is another hyperparameter. The message is then fed through the gate activation layer, which adds non-linearity while preserving equivariance by applying separate non-linearities to the $l = 0$ and $l > 0$ components.

Next, the non-linear message is passed through a second convolution layer that produces an output embedding size of $(n_e, (l_{max}+1)^2, d_{attn\_value}*N_{heads})$, which is then be reshaped into $(n_e, (l_{max}+1)^2, N_{heads}, d_{attn\_value})$. For each edge surrounding the target node, a set of $d_{attn\_alpha}$ attention weights are then generated for each attention head, with the total number of heads being $N_{heads}$. This is used to generate the output vector alpha, which is reshaped along $N_{heads}$ and multiplied with

the reshaped message embedding. Finally, the messages from neighboring are reshaped, rotated back, and aggregated onto the target node, before being projected back into the shape $(N_n, (l_{max} + 1)^2, d_{sphere})$.

This is finally passed into a feed-forward network consisting of two linear layers that mixes features of the same $l$ together. The hidden dimensions used in the feed-forward network is given by the hyperparameter $d_{ffn}$. The edge update block is similar to the node update block, except there is only one convolution layer and no attention required. Between the blocks, layer normalisation is also applied, and similar to EquiformerV2, we also normalised the $l = 0$ features separately from the $l > 0$ features. The final predicted output is passed into the Wigner Eckart layer to be reconstructed into Hamiltonian blocks.

## A.1 WIGNER ECKART LAYER

Our implementation of this layer is similar to that found in Gong et al. (2023). A Hamiltonian block representing the interaction between atom $i$ and $j$ consists can be split into different sub-blocks representing interactions between orbital 1 of order $l_1$ and orbital 2 of degree $l_2$ as shown in Fig. 1. Each block of size $(2l_1 + 1) \times (2l_2 + 1)$, is an equivariant tensor that comes from the tensor product $l_1 \otimes l_2$ for every pair of interacting orbitals in atom $i$ and atom $j$. For example, an interaction block between a $p$ orbital ($l = 1$) and a $d$ orbital ($l = 2$) has $(2 \times 1 + 1) \times (2 \times 2 + 1) = 3 \times 5$ elements. We refer to this form as the uncoupled basis representation of the Hamiltonian block.

Before training, the Wigner-Eckart layer converts the Hamiltonian data from uncoupled basis to coupled basis using Clebsch Gordan coefficients.

$$l_1 \otimes l_2 = |l_1 - l_2| \oplus ... \oplus (l_1 + l_2)$$

$l_1 \otimes l_2$ is now represented by a direct sum of coupled sub-spaces with order ranging from $|l_1 - l_2|$ to $(l_1 + l_2)$. This is repeated for every possible type of orbital interaction between the atoms, to obtain the final direct sum of all the sub-spaces needed for the network to reconstruct the Hamiltonian.

The largest possible $(l_1 + l_2)$ value determines the minimum $l_{max}$ hyperparameter needed for the embeddings of the equivariant model. In the case of $HfO_2$ in this paper, which uses the SZV basis set, the highest order orbital is the $d$ orbital of Hf. This means that the largest possible $(l_1 + l_2)$ comes from the interaction between two Hf $d$ orbitals, and the $l_{max}$ needs to be at least equal to $(2+2) = 4$. This allows the predicted output of the model to be converted back into the full uncoupled basis using the same layer, and reassembled into the full Hamiltonian matrix during inference.

## A.2 LOSS COMPUTATION DURING TRAINING AND INFERENCE

For all experiments, a minor difference from the procedures reported in Yu et al. (2023b) and Schütt et al. (2019) is that we use the Mean Squared Error (MSE) of the full target vectors in the coupled space to compute the fitting and validation loss during training:

$$\mathcal{L}_{MSE}(x_i, \hat{x}_i) = \frac{1}{N} \sum_{i=1}^{N} (x_i - \hat{x}_i)^2$$

Where $x$ and $\hat{x}$ are the flattened targets (orbital blocks). These targets are padded with zeros to ensure that those of different orbital interactions have the same dimensions. To avoid re-shaping the predictions within every epoch, the loss computed during training includes this padding. This is procedure was also used in DeepH E3 (Gong et al. (2023)). However, the final reported loss in Table 6 uses the Mean Absolute Error (MAE) after converting the output and label tensors back into uncoupled space and reconstructing the Hamiltonian blocks.

$$\mathcal{L}_{MAE}(x_i, \hat{x}_i) = \frac{1}{N} \sum_{i=1}^{N} |x_i - \hat{x}_i|$$

The padding is omitted during the reconstruction process, and all the elements in the blocks where the final MAE is computed from represent orbital interactions that exist in the label Hamiltonian.

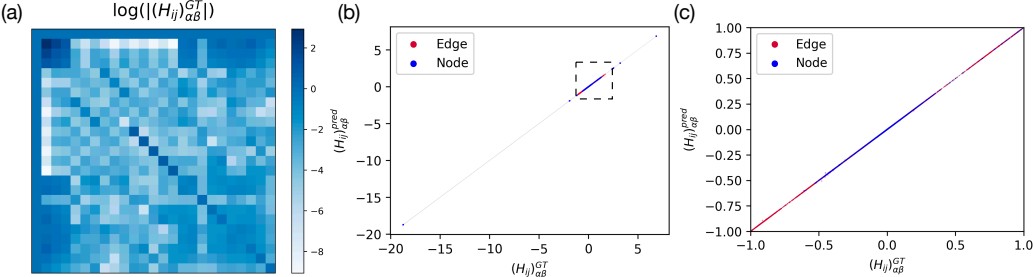

Figure 8: Training the MD17 Hamiltonian matrices of $H_2O$ molecules. (a) Elements of the hamiltonian matrix for a sample $H_2O$ molecule in the dataset. (b) Fitting of the predicted elements of the Hamiltonian matrix against the labels, after randomized downsampling to every 100 data points. Data points closer to the grey line indicate better agreement. (c) Magnification of fit around [-1, 1].

## B    PERFORMANCE ON SMALL MOLECULES ($H_2O$)

As our work combines and adapts existing methodology to a new, application-oriented dataset, we first ensure that our adaptation of the networks introduced by Liao et al. (2023) and Wang et al. (2024a) can achieve reasonable results on an existing dataset. We select the dataset of $H_2O$ molecules in MD17 Schütt et al. (2019). The hyperparameters which determine the network size are $d_{sphere}$ and $d_{attn\_hidden}$ (**Table 8**). In **Table 5** we explore the effect of these parameters under early-stopping conditions. We fix the number of MP layers to 2 in all cases, lower than was used by similar equivariant GNNs (Unke et al. (2021), Yu et al. (2023b)). Channel dimensions of 32 and above meet the loss defined for the stopping criteria, with larger dimensions reaching it in fewer epochs.

We select a dimension of 64, and train the network under the conditions specified in **Table 8**. In **Table 6** we compare the final Mean Absolute Error (MAE) loss with similar networks in the literature. Our implementation can achieve a prediction accuracy of $\sim 100 \ \mu E_h$, which is within an order of magnitude of other equivariant networks while using a smaller network (#MP layers).

| Dim. | # Epochs | Training Molecules | | Testing Molecules | |
|---|---|---|---|---|---|
| | | $\epsilon_{\mathbf{node}} \ [\times 10^{-6} E_h]$ | $\epsilon_{\mathbf{edge}} \ [\times 10^{-6} E_h]$ | $\epsilon_{\mathbf{node}} \ [\times 10^{-3} E_h]$ | $\epsilon_{\mathbf{edge}} \ [\times 10^{-3} E_h]$ |
| 16/16 | 50,000* | 0.20* | 2.81* | 0.1498* | 0.4776* |
| 32/32 | 10,847 | 0.54 | 0.43 | 0.1716 | 0.1866 |
| 64/64 | 6,761 | 0.36 | 0.32 | 0.1322 | 0.1534 |
| 128/128 | 2,015* | 1.08* | 1.01* | 0.1878* | 0.2776* |

Table 5: Effect of the channel dimension used to train the network on $H_2O$. For this study we remove 1499 molecules from the dataset (4999 total molecules), and use 500 for training (single-batch), 500 for validation, and 499 for testing. For fair comparison between the different network sizes, the trainings were subject to an early-stopping criteria of $\epsilon_{MSE} = 1 \times 10^{-6}$ of the validation set loss. Training losses are reported in MSE while testing losses are reported in MAE. The '*' indicates that the error on the validation set did not reach the stopping criteria, and training instead finished when $lr=\epsilon_{MSE}$ (128/128) or when the # epochs reached 50,000.

| Network | MAE **H** $[meV]$ | MAE **H** $[\times 10^{-6} E_h]$ |
|---|---|---|
| PhiSNet | 0.47 | 18 |
| QHNet | 0.29 | 11 |
| **This work** | 2.7 | 100 |

Table 6: Mean Absolute Error (MAE) for predicted Hamiltonian matrices reported for PhiSNet (Unke et al. (2021)), and QHNet (Yu et al. (2023b)) on the MD17 $H_2O$ dataset, taken from the respective publications. The result for our adapted EquiformerV2 network for electronic property prediction uses 500 molecules for testing, 500 for validation, 2500 for testing. Note that we use 2 layers, as opposed to PhiSNet (4) and QHnet (5).

## C  Augmenting graph partitions with virtual nodes

Below we detail the procedure to partition the full graph $\mathcal{G}$, described by the set of vertices $\mathcal{V}$ and edges $\mathcal{E}$, into a set of slices $\{\mathcal{G}_1 \ \ldots \ \mathcal{G}_N\}$ which are augmented by virtual nodes and edges.

---

**Algorithm 1:** Augmented partitioning approach

1  Graph $\mathcal{G}(\mathcal{V}, \mathcal{E})$ Slice location $[x_1 \ \ldots \ x_N]$ Set of subgraphs $\{\mathcal{G}_1 \ \ldots \ \mathcal{G}_N\}$ Numbers labeled nodes $[n_1^n \ \ldots \ n_N^n]$ Numbers labeled edges $[n_1^e \ \ldots \ n_N^e]$

2  **for** $i \leftarrow 1$ **to** $N$ **do**
3     $\mathcal{V}_i \leftarrow [\,]$;
4     $n_i^n = 0$;
5     **for** $v \in \mathcal{V}$ **do**
6        **if** $v.x \in [x_i, x_{i+1}]$ **then**
7           $\mathcal{V}_i$.append(v);
8           $n_i^n \mathrel{+}= 1$;
9        **end**
10    **end**
11    $\mathcal{E}_i \leftarrow [\,]$;
12    $n_i^e = 0$;
13    **for** $v_1 \in \mathcal{V}_i$ **do**
14       **for** $v_2 \in \mathcal{V}_i$ **do**
15          **if** $v_1 \rightarrow v_2 \in \mathcal{E}$ **then**
16             $\mathcal{E}_i$.append($v_1 \rightarrow v_2$);
17             $n_i^e \mathrel{+}= 1$;
18          **end**
19       **end**
20    **end**
21    **for** $v_1 \in \mathcal{V}_i$ **do**
22       **for** $v_2 \in \mathcal{V} \setminus \mathcal{V}_i$ **do**
23          **if** $v_2 \rightarrow v_1 \in \mathcal{E}$ **then**
24             $\mathcal{V}_i$.append($v_2$);
25             $\mathcal{E}_i$.append($v_2 \rightarrow v_1$);
26          **end**
27       **end**
28    **end**
29    $\mathcal{G}_i(\mathcal{V}_i, \mathcal{E}_i)$;
30  **end**

---

The number of labeled nodes ($n_i^n$) and the number of labeled edges ($n_i^e$) are collected and passed to the training functions, which then mask the remainder of the outputs (the virtual nodes and edge outputs) while computing the loss.

## D  Augmented partition approach with two MP layers

| $t_{slice}$ [Å] | $N_t$ | # Epochs | $\epsilon_{\mathbf{node}}$ | $\sigma_{\mathbf{node}}$ | $\epsilon_{\mathbf{edge}}$ | $\sigma_{\mathbf{edge}}$ |
|---|---|---|---|---|---|---|
| $\sim 2$ | 27 | 16,437 | 8.24 | 21.35 | 0.24 | 0.39 |
| $\sim 3$ | 28 | 13,461 | 8.60 | 21.01 | 0.24 | 0.43 |
| $\sim 4$ | 14 | 14,089 | 6.34 | 15.06 | 0.24 | 0.38 |
| $\sim 6$ | 9 | 11,670 | 4.46 | 11.50 | 0.24 | 0.39 |

Table 7: Ablation studies exploring the impact of slice length $t_{slice}$ on the prediction accuracy with 2 MP layers. Values are reported in $\epsilon_n [mE_h]$ and $\sigma_n [\mu E_h]$.

Using 2 MP layers rather than one heavily degrades $\epsilon_{node}$ (**Table 3**). This occurs because each slice is augmented with only one layer of virtual connections, so the network does not have a correct

representation of the 2-hop neighborhood. Propagation of this incorrect information into the graph thus results in a degraded $\epsilon_{node}$, which is more sensitive to the extended environment that $\epsilon_{edge}$. The error in this case minimally depends on the outgoing edge (albeit uniformly high). We note that the error with 2 MP layers shows a dependence on $t_{slice}$ (**Table 9**) since larger slices require fewer virtual connections outside the partition.

# E HYPERPARAMETERS

We use the hyperparameters shown in Table 8 to train the $H_2O$ benchmark and custom $HfO_2$ dataset. The ReduceLRonPlaeau scheduler decreases the learning rate by the decay factor when it does not detect a further decrease in validation loss within the decay patience $t_{patience}$. The threshold refers to the sensitivity of the scheduler to changes in validation loss. Once the minimum learning rate is reached, the training stops. The meaning of the hyperparameters are explained in Appendix A. Note that weight decay is not implemented in our study for all cases.

| Hyper-parameters | HfO$_2$/PtGe dataset | MD17 dataset |
|---|---|---|
| Optimizer | Adam | Adam |
| Precision | single (f32) | double (f64) |
| Scheduler | ReduceLROnPlateau | ReduceLROnPlateau |
| Initial learning rate | $1 \times 10^{-4}$ | $1 \times 10^{-4}$ |
| Minimum learning rate | $1 \times 10^{-5}$ | $1 \times 10^{-10}$ |
| Decay patience t$_{patience}$ | 500 | 50 |
| Decay factor | 0.5 | 0.5 |
| Threshold | $1 \times 10^{-3}$ | $1 \times 10^{-5}$ |
| Interaction distance (Å) | 8.0 | – |
| Maximum degree $L_{max}$ | 4 | 4 |
| Maximum order $M_{max}$ | 4 | 4 |
| Number of Message Passing Layers | 1 | 2 |
| Number of spherical channels $d_{sphere}$ | 16 | 64 |
| $f_{ij}^{(L)}$ dimension $d_{attn\_hidden}$ | 16 | 64 |
| Number of attention heads $N_h$ | 2 | 2 |
| $f_{ij}^{(0)}$ dimension $d_{attn\_alpha}$ | 16 | 32 |
| Value dimension $d_{attn\_value}$ | 16 | 32 |
| Hidden dimension in feed forward networks $d_{ffn}$ | 64 | 64 |

Table 8: Hyper-parameters used for HfO2, PtGe and MD17 data.

## E.1 HYPERPARAMETER STUDY

We conducted hyperparameter tuning by training using the full training structure (structure 2), divided into 18 slices of $\sim 3$ Å thickness. The validation slice used for the scheduler during training was taken from the validation structure, and is centered at the location 26.5 Å. The rest of the hyperparameters follow the values in Table 8. Note that since we are tuning the hyperparameters, we evaluate the trained models with different hyperparameters on the full validation structure (structure 1). The test structure remains unseen throughout this process.

From the table, it is clear that the hyperparameters $d_{sphere}$ , $d_{attn\_value}$, $d_{attn\_hidden}$ and $d_{attn\_alpha}$ can all be reduced to 16 with little tradeoff in accuracy. Cutting down on the number of parameters allows us to drastically minimise the memory consumption, allowing large graphs to fit into GPU memory during training.

| $d_{sphere}$, $d_{attn\_hidden}$ | $d_{attn\_value}$, $d_{attn\_alpha}$ | Parameters | Epochs | $\epsilon_{\mathbf{n}}[mE_h]$ | $\sigma_{\mathbf{n}}$ | $\epsilon_{\mathbf{e}}[mE_h]$ | $\sigma_{\mathbf{e}}$ |
|---|---|---|---|---|---|---|---|
| 4 | 32 | 52,572 | 31,940 | 2.82 | 5.72 | 0.36 | 0.84 |
| 8 | 32 | 125,324 | 29,002 | 2.54 | 5.73 | 0.18 | 0.35 |
| 16 | 16 | 273,644 | 19,106 | 2.39 | 5.31 | 0.16 | 0.33 |
| 16 | 32 | 335,436 | 16,857 | 2.46 | 5.84 | 0.19 | 0.35 |
| 32 | 32 | 1,014,092 | 14,833 | 2.51 | 5.83 | 0.19 | 0.37 |
| 64 | 32 | 3,405,132 | 11,000 | 2.47 | 5.83 | 0.19 | 0.37 |

Table 9: Table of Hyperparameters, tested on the validation structure (structure 1)

# F  AMORPHOUS HAFNIUM DIOXIDE (A-HfO$_2$) TRAINING DATA

## F.1  DATASET GENERATION

Atomic structures corresponding to materials in the amorphous phase can be produced through melt-quench Urquiza et al. (2021), seed-and-coordinate Youn et al. (2014), or 'decorate and relax' Tafen & Drabold (2003) approaches. To accurately reproduce long-range structural disorder, the structures used must be large enough to avoid the creation of wavefunctions which repeat over periodic boundaries.

The first step to computing the electronic properties of a-HfO$_2$ is to generate the atomic structure of the material in its amorphous phase. To accurately capture the structural motifs underlying this phase and a realistic range of atomic coordination (eg, neighboring Oxygens for each Hafnium, and vice versa), we start from the crystalline m-HfO$_2$ phase and perform melt-quench processes using Molecular Dynamics (MD), following a similar procedure as the ones described in Refs. Kaniselvan et al. (2023); Urquiza et al. (2021). We generate 3 independent structures of a-HfO$_2$ using the QuantumATK toolkit Søren Smidstrup et al. (2020). As the first step, we run an NVT simulation with the Langevin thermostat at 3000 K for 50 ps with a step size of 1 fs. We use the MTP-HfO$_2$-2022 potential, provided by the software. Next, we run an NPT simulation for 300 ps (and the same 1 fs step size), with an initial reservoir temperature of 3000K and a final temperature of 300K, for a cooling rate of 9K/ps. Finally, we anneal structure evolve at 300K for 50 ps, using the same NVT Langevin thermostat as for the melting.

We then perform a structural relaxation with CP2K code Kühne et al. (2020) to correct for any discrepancies between the relaxed bond lengths attained with the force field used for MD, and those obtained with DFT. Due to the computational cost of using a more complete DZVP basis set Vande-Vondele & Hutter (2007), we use a simpler SZV basis VandeVondele & Hutter (2007) which uses 4 basis functions per Oxygen atom and 10 basis functions per Hafnium atom. The plane-wave cutoff is set to 500 Ry, while a cutoff of 60 Ry is used for mapping the Gaussian-type orbitals onto the grid. We use the PBE functional for the exchange-correlation energy Perdew et al. (1997). To accurately capture the band gap of a-HfO$_2$, we applied the Hubbard correction Anisimov et al. (1991) of U = 7 eV to the 3d orbital of Ti and the Hubbard correction of U = 10 eV to the 2p orbital of O.

## F.2  ATOMIC BONDING ENVIRONMENTS IN THE AMORPHOUS PHASE

In **Fig. 9** we plot the O-coordination of each Hf atom and the radial distribution function $g(r)$ (where r is inter-atomic distance) for each of the three structures. The distribution in the coordination and dispersion of the peaks in $g(r)$ indicates the amorphous nature of the three structures. Variations between them appear as perturbations in these two quantities. To gain more insights on how different are the structures, we additionally plot the spatially-resolved O-coordination of Hf atoms along the longest, x coordinate for the three structures, as well as the distributions of outliers (Hf atoms with very low and very high O-coordination) in three-dimensional space. It is evident that these outliers are situated at different locations in different structures, demonstrating a significant degree of dissimilarity among the structures.

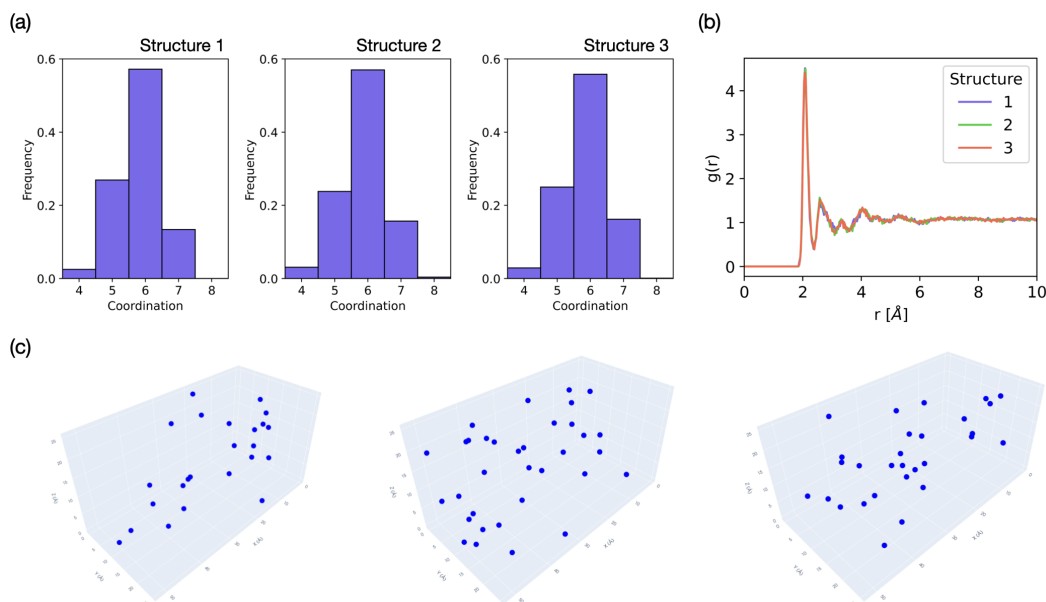

Figure 9: a) O-coordination of Hf atoms (number of O atoms bonding a Hf atom) for each of the generated structures, showing a distribution around a coordination number of 6, and variation between the structures. b) The radial distribution function ($g(r) = \frac{dn(r)}{dr} \frac{V_{domain}}{4\pi r^2 N_{atoms}}$), where $n(r)$ is the number of atoms with distance $r$ between them for the structures 1-3. (c) Spatial distribution of coordination outliers (Hf atoms with O-coordination equal to 8 or 4) for the three structures, which are an indicator of the uniqueness of the three structures.

### F.3 ENERGY EIGENVALUES

Due to the large cell sizes of the a-HfO$_2$ structures, all necessary energetic information is contained within the $\Gamma$ point (where the wavevectors $k_x = k_y = k_z = 0$). The energies at this location can be computed by directly diagonalizing $\mathbf{H}$. Amorphous a-HfO$_2$ structures corresponding to a realistic distribution of bond lengths should then produce an energy bandgap. We show the distribution of energy eigenvalues for the three structures in **Table 1** in **Fig. 10**, at different values of $r_{cut}$. In the second row, we zoom into the range of eigenvalues around the energy bandgap, which is defined by the transition between occupied and unoccupied electronic states (the Fermi level $E_F = \sim 0.3 E_h$ in all cases). Values of $r_{cut} \geq 8\text{Å}$ create no noticeable difference on the eigenvalue spectra. Note that the value of $r_{cut} = \infty$ corresponds to the case where no nonzero values were filtered from $\mathbf{H}^{GT}$.

## G COMPUTE ENVIRONMENT AND RUNTIME COMPARISONS

The training is performed with PyTorch Distributed Data Parallel ( Li et al. (2020)), where the graph partitions (slices) can be distributed between GPUs.

### G.1 MEMORY CONSUMPTION OF THE FULL GRAPH

During the training of the full graph model, the peak memory consumption observed was 61.68 GiB on a single NVIDIA A100 GPU. Most of the consumption does not stem from the network and the structure but from the additional memory needed for the convolution operations.

### G.2 COMPUTATIONAL IMPROVEMENT WITH PARTITIONING APPROACH

In **Fig. 11**, we show the decrease in time per epoch and resulting speedup when using the *augmented partitioning* approach introduced in **Section 3.2**. Since the partitions are independent, the only communication involved in every epoch is a collective to inform each GPU/rank of the loss of each other rank. The time per epoch thus decreases uniformly with the number of slices ($N_t$) used.

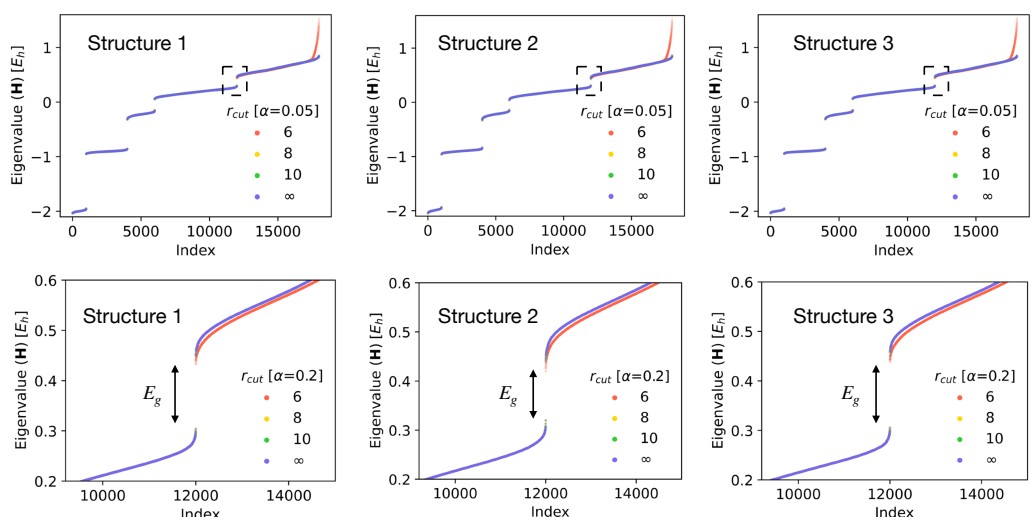

Figure 10: Eigenvalues of the ground-truth Hamiltonian matrix, showing (left) the full eigenvalue spectrum and (right three) zoomed in around the bandgap, for the three structures, in the order of their appearance in **Table 1**.

Despite the independence of each batch and the minimal communication per epoch, the scaling is not perfectly linear. The deviation from an ideal speedup can be attributed to two factors:

- Load imbalance: The partitioning approach was designed to leverage the periodicity in the $y$- and $z$- direction within a straightforward implementation. However, it is not ideal in terms of the number of cuts/number of virtual nodes/edges required, resulting in a slightly different amount of work per rank which leads to an observable load imbalance at higher $N_t$. This effect can be seen in the allocated memory per partition (**Fig. 11(c)**). We note that the *augmented partitioning* method can be used with any standard graph-partitioning algorithm.

- Computational overhead of the virtual nodes and edges: Individual nodes and edges of the graph can be repeated in labeled and virtual node lists. Treating the replicas introduces additional computational cost while training the network, which increases with $N_t$. This overhead is maximum with the use of very small slices (large $N_t$), thus introducing a trade-off between parallelism and time per epoch.

### G.3 $H_2O$ VS $HfO_2$ RUNTIMES

In **Section 4.5**, we make a comparison between the computational cost of computing the Hamiltonian for an $H_2O$ molecule and the $HfO_2$ structure. However, the Hamiltonians for the $H_2O$ molecules in the MD17 dataset were computed by Schütt et al. (2019) using different compute infrastructure. To approximate the cost of generating such a dataset under the same computational conditions, we set up CP2K simulations with a Double-$\zeta$ Valence Polarized (DZVP) basis, which includes a similar set of valence and polarization functions as the def2_SVP basis used with the ORCA code (Neese (2011)) to generate the $H_2O$ Hamiltonians for the MD17 dataset. A minor difference is that the def2_SVP basis includes an extra s-type orbital on Oxygen (14 total per Oxygen atom). Under these conditions, the computation time per $H_2O$ molecule was 7s, when run on 12 nodes with 12-core Intel Xeon E5-2680 CPUs and NVIDIA P100 GPU (the Piz Daint supercomputer), resulting in a total of 0.04 node hours. The $HfO_2$ structures requires 3.65 node hours in the same compute environment (but distributed to 27 nodes). The difference, omitting scaling behavior, is roughly $\sim 100\times$.

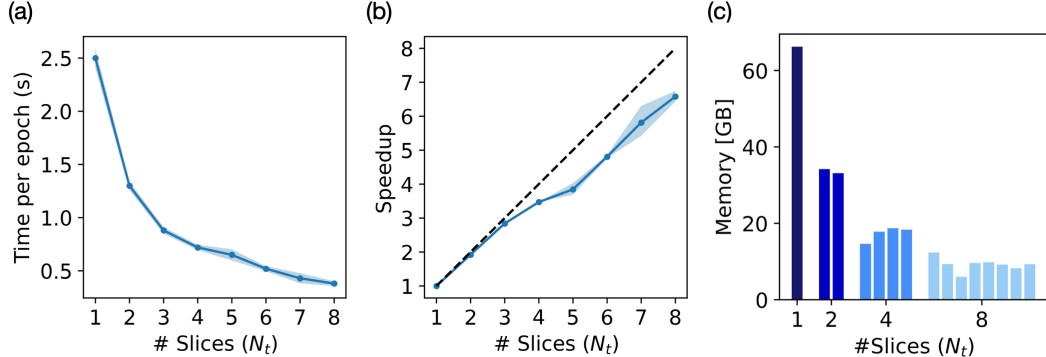

Figure 11: (a) Time per epoch and (b) speedup resulting from the use of increasing numbers of slices $N_t$. Median values are shown, while the error bands are one standard deviation. Experiments were run on NVIDIA A100 GPUs with # ranks set to $N_t$. Measurements are only shown up to 8 slices/8 GPUs due to limitations in available compute resources at the time of submission. The fill-between indicates the range in runtime over the first 30 minutes of training. The dashed black line corresponds to the ideal speedup, in which case the use of $N_t$ slices would enable an $N_t\times$ speedup in the runtime per epoch. (c) Measured peak memory consumption as a function of the number of partitions, where each bar corresponds to a different GPU. Variation in memory consumption between GPUs at each individual value of $N_t$ translates to load imbalance, which correlates with the deviation from ideal scaling shown in (b).

## H COMPARISONS BETWEEN DFT AND PREDICTED HAMILTONIANS

In **Figure 12**, we plot the MAE error as a function of different interactions between the orbital basis of the a-HfO$_2$ test structure. The data is plotted in log scale to magnify the asymmetry resulting from the separate training of the two-way edges between labeled nodes in the graph.

In **Fig. 13**, we plot the comparison between $H^{pred}$ and $H^{GT}$ (as shown in the main text) for three separate cases: (1) using the upper triangle, (2) the lower triangle, and (3) the symmetrized $H^{pred}$. We also zoom in around the bandgap in the second row. In all three cases, $H^{pred}$ is unchanged, indicating that the small asymmetry in the matrices caused by the existence of separate forward/backward edges between atoms has a minimal effect. In the third row, we show the error in log scale as a function of the eigenvalue index, sorted in the same order as shown in the first row. The error is largest around the band edges, where orbital coupling is most significant.

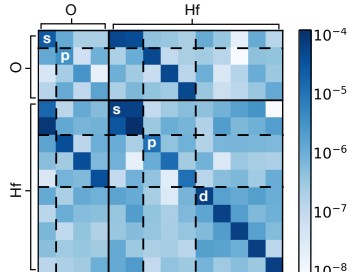

Figure 12: Average $\epsilon_{MAE}$ between $(H_{ij})^{GT}$ and $(H_{ij})^{pred}$ for specific inter-atomic orbital coupling. The data is shown in log scale to magnify asymmetry in the error.

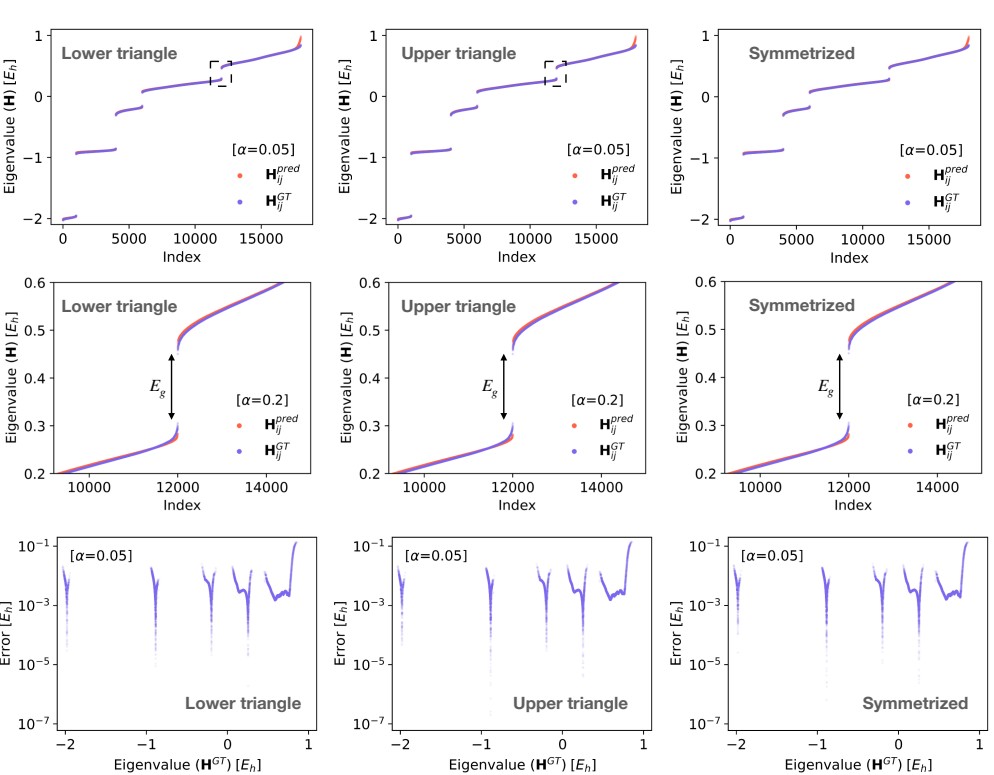

Figure 13: Comparison between $\boldsymbol{H}^{GT}$ and $\boldsymbol{H}^{pred}$, using the upper/lower triangle or symmeterized version of $\boldsymbol{H}^{pred}$. The center row shows the same plots zoomed in around the bandgap. The last row shows the difference in the eigenvalue spectra. The structures appear in the order of **Table 1**.

| Structures | | Epochs | $\epsilon_\mathbf{n} \, [mE_h]$ | $\sigma_\mathbf{n} \, [\mu E_h]$ | $\epsilon_\mathbf{e} \, [mE_h]$ | $\sigma_\mathbf{e} \, [\mu E_h]$ | $\epsilon_\mathbf{tot} \, [mE_h]$ | $\epsilon_\mathbf{tot} \, [meV]$ |
|---|---|---|---|---|---|---|---|---|
| Training | Testing | | | | | | | |
| 0 | 3 | 15675 | 2.29 | 5.09 | 0.20 | 0.38 | 0.22 | 5.87 |
| 0, 2 | 3 | 48356 | 2.22 | 4.99 | 0.12 | 0.25 | 0.13 | 3.55 |

Table 10: Comparison of models trained on one HfO$_2$ structure (structure 2) vs two HfO$_2$ structures (0 and 2). Both are tested on a fully unseen structure (3), showing improved accuracy from 5.87 to 3.55 $meV$.

| Oxygen vacancies | | $\epsilon_\mathbf{n} \, [mE_h]$ | $\sigma_\mathbf{n} \, [\mu E_h]$ | $\epsilon_\mathbf{e} \, [mE_h]$ | $\sigma_\mathbf{e} \, [\mu E_h]$ |
|---|---|---|---|---|---|
| Training set | Testing set | | | | |
| 5% | 5% | 2.44 | 5.06 | 0.16 | 0.31 |
| 5% | 10% | 2.58 | 5.23 | 0.18 | 0.33 |
| 5% | 15% | 2.50 | 4.96 | 0.17 | 0.33 |
| 10% | 5% | 2.48 | 5.12 | 0.18 | 0.33 |
| 10% | 10% | 2.50 | 4.96 | 0.17 | 0.33 |
| 10% | 15% | 2.60 | 4.89 | 0.18 | 0.33 |
| 15% | 5% | 2.94 | 6.45 | 0.16 | 0.34 |
| 15% | 10% | 2.52 | 5.01 | 0.16 | 0.34 |
| 15% | 15% | 2.52 | 4.69 | 0.16 | 0.31 |

Table 11: HfO$_2$ models trained and tested with different stoichiometry using augmented partitioning. 18 slices of each structure, each 3 Å thick, was used for training. The training method is identical to the one used to obtain Table 4. Models trained on vacancies 5%, 10 % and 15% vacancies are tested on test structures with vacancies ranging from 5% to 15%.

# I ADDITIONAL TESTS

## I.1 USING TWO HfO$_2$ STRUCTURES FOR TRAINING

While training on a single structure is sufficient to achieve the results in **Fig. 6**, increasing the quantity of training data further leads to improved prediction accuracy. To demonstrate this, we train using *augmented partitioning* on two structures, including structure 2 from **Table 1** as well as a newly generated structure "0" of similar size (3000 atoms). Training was performed with 18 slices of 3 Å thickness taken from both structures, validated on structure 1, and tested on structure 3. The results are compared against the previous model trained on one structure in Table 10, showing that the prediction accuracy can be improved from 5.87 $meV$ to 3.55 $meV$ by using 2× the number of slices for training.

## I.2 SUB-STOICHIOMETRIC HAFNIUM OXIDE

Amorphous HfO$_2$ often exists in a sub-stoichiometric form (HfO$_x$), which can be interpreted as the presence of oxygen vacancies. In this section we evaluate whether our model can extend to train and predict such defective structures.

To do this, we create a dataset for sub-stoichiometric HfO$_x$ structures by introducing randomly distributed oxygen vacancies into the original, pristine HfO$_2$ structures. The sub-stoichiometric structures are generated for x = 1.9, 1.8, and 1.7 (corresponding to vacancy concentrations of 5%, 10%, and 15 %, respectively). Vacancies are treated as ghost atoms (atoms with no orbitals, but with a basis set defined at their locations), to mitigate the basis set superposition error Senent & Wilson (2001), a known problem related to localized basis sets. More precisely, by treating vacancies as ghost atoms, one prevents the excessive borrowing of the basis sets from neighboring atoms by the vacancy, which improves the accuracy of the predicted electronic properties. These ghost atoms are assigned an atomic number of 0. The training and testing approach is similar to the one used to obtain Table 4, except that now oxygen vacancies are considered. For all experiments, 18 slices (3 Å thick) were used.

The results are summarized in Table 11. The $\epsilon_n$ and $\epsilon_e$ values across different experiments lie within a small range (2.50-2.90 $mE_h$ and 0.16-10.18 $mE_h$ respectively), showing that the network generalizes well to structures of different vacancies, regardless of which vacancy configuration it was

| Training method method | Oxygen vacancies Testing set | $\epsilon_{\mathbf{n}} [mE_h]$ | $\sigma_{\mathbf{n}} [\mu E_h]$ | $\epsilon_{\mathbf{e}} [mE_h]$ | $\sigma_{\mathbf{e}} [\mu E_h]$ |
|---|---|---|---|---|---|
| partitioned | 5% | 2.94 | 6.45 | 0.16 | 0.34 |
| partitioned | 10% | 2.52 | 5.01 | 0.16 | 0.34 |
| partitioned | 15% | 2.52 | 4.69 | 0.16 | 0.31 |
| full | 5% | 2.96 | 6.07 | 0.19 | 0.38 |
| full | 10% | 2.67 | 5.18 | 0.18 | 0.36 |
| full | 15% | 2.64 | 4.83 | 0.17 | 0.35 |

Table 12: Comparison between full graph training and the augmented partitioning training using the same $HfO_2$ structure with 15% vacancies. Models are tested on structures with vacancies ranging from 5% to 15%.

| Material | Cutoff $[\mathring{A}]$ | $\epsilon_{\mathbf{n}} [mE_h]$ | $\sigma_{\mathbf{n}} [\mu E_h]$ | $\epsilon_{\mathbf{e}} [mE_h]$ | $\sigma_{\mathbf{e}} [\mu E_h]$ | $\epsilon_{\mathbf{tot}} [mE_h]$ | $\epsilon_{\mathbf{tot}} [meV]$ |
|---|---|---|---|---|---|---|---|
| Crystalline $HfO_2$ | 8 | 0.01 | 0.02 | 0.04 | 0.07 | 0.04 | 1.17 |
| Amorphous $HfO_2$ | 8 | 2.29 | 5.09 | 0.20 | 0.36 | 0.22 | 5.87 |
| Amorphous PtGe | 16 | 0.87 | 1.43 | 0.05 | 0.10 | 0.05 | 1.43 |

Table 13: Summary of models trained on crystalline $HfO_2$, amorphous $HfO_2$ and amorphous PtGe materials, respectively. The $HfO_2$ model was trained on different slices of the same crystalline structures. On the other hand, the PtGe model was trained on a single 5 Å slice of structure 1 and tested on on a fully unseen structure 2.

trained on. To demonstrate that the *augmented partitioning* approach similarly does not affect accuracy for sub-stoichiometric $HfO_x$, we also perform full graph training using structure 2 with 15% vacancies, and compare with the *augmented partitioning* approach in Table 12. The minimal difference in $\epsilon_n$ and $\epsilon_e$ values between full and partitioned approaches indicates that both approaches generalize equally well to different stoichiometry. These values are also close to that of stoichiometric $HfO_2$ in Table 4, demonstrating that the augmented partitioning approach can also be applied even in the case of more realistic sub-stoichiometric structures.

### I.3 CRYSTALLINE HAFNIUM OXIDE

Crystalline materials contain highly regular atomic environments, and are thus a natural extension of this approach. Although such a large unit cell is not required for a crystal, we nevertheless generate a single crystalline $HfO_2$ structure (in its monoclinic phase) containing 3000 atoms for comparison. The model was trained on a single slice taken from x = 0 Å, validated on a slice from x = 15 Å, and tested on an unseen slice from x = 20 Å. Results in Table 13 show that a high accuracy close to sub-$meV$ can be achieved.

### I.4 SECOND MATERIAL EXAMPLE: PTGE

Finally, we test if the model and training approach can be used for different material systems. We consider the example of two amorphous PtGe structures, labelled 1 and 2, each containing 2688 atoms. These structures were similarly generated in CP2K. A larger cutoff radius of 16 Å was chosen due to the larger spacing between atoms. The model was trained on a single 5 Å slice taken from x = 10 Å and validated on another slice at x = 20 Å, both from structure 1. The trained model was then tested on a full unseen structure (structure 2), with results shown in Table 13. The final obtained error for all 2688 atoms and 2,148,055 edges is 1.43 $meV$, much lower than that of $HfO_2$, despite the lower amount of training data. This demonstrates the generalizability and robustness of

| Cutoff $[\mathring{A}]$ | $\epsilon_{\mathbf{n}} [mE_h]$ | $\sigma_{\mathbf{n}} [\mu E_h]$ | $\epsilon_{\mathbf{e}} [mE_h]$ | $\sigma_{\mathbf{e}} [\mu E_h]$ | $\epsilon_{\mathbf{tot}} [mE_h]$ | $\epsilon_{\mathbf{tot}} [meV]$ |
|---|---|---|---|---|---|---|
| 6 | 0.87 | 1.43 | 0.15 | 0.19 | 0.17 | 4.60 |
| 8 | 0.87 | 1.42 | 0.09 | 0.16 | 0.10 | 2.73 |
| 16 | 0.87 | 1.43 | 0.05 | 0.10 | 0.05 | 1.43 |

Table 14: Prediction accuracy of model on amorphous PtGe material with different $r_{cut}$

the augmented partitioning approach when applied to different material dataset with a much larger cutoff radius.

We perform a similar study on the cutoff radius and connectivity of the PtGe, through both eigenvalue analysis and the convergence study of cutoff-radii, with results shown in Table 14 and Figure 14. Increasing the cutoff radius once again increases the overall prediction accuracy of the trained model, with the improvement especially noticeable at the edges.

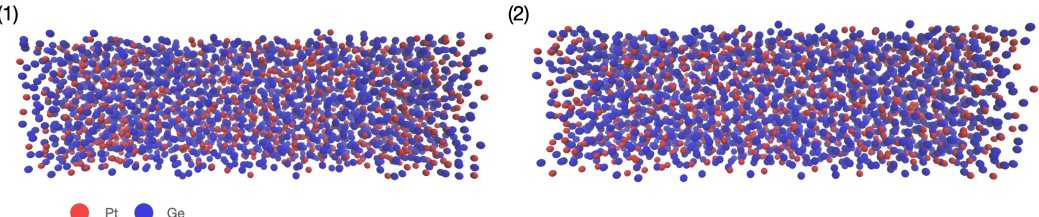

Figure 14: Atomic coordinates of the two amorphous PtGe structures.

## J  THEORETICAL JUSTIFICATION FOR AUGMENTED PARTITIONING APPROACH

The augmented partitioning approach relies on the correction of the partitioned sub-graphs through the introduction of virtual nodes and edges, such that partitioned training fully resembles full graph training. Here, we provide the theoretical foundations upon which our approach is built.

We start by recalling the operation of a message passing neural network trained on a full graph. During training, the nodes are updated through the aggregation of messages from connected neighbors. More specifically, in the first layer of equivariant message passing networks, the $n_i'$ inputs to the aggregation function are simply the atomic numbers $Z_i$ and $Z_j$ (embedded in a tensor) of atom $i$ and neighbor $j$, their scalar distances $|r_{ji}|$ (expanded in the Gaussian basis), and the normalized vector $\hat{r}_{ji}$ indicating the orientation of the edges (embedded within the rotation and counter-rotation operations)

$$n_i' = \sum_{j \in \mathcal{N}(i)} \Phi_j(Z_i, Z_j, |r_{ji}|, \hat{r}_{ji}). \tag{1}$$

In Eq. (1), $\mathcal{N}(i)$ is the neighbor list of atom $i$ and $\Phi_i$ is a learnable function that encompasses all the operations of our equivariant network (convolution, gate activation, attention weights). The sum of these functions over all neighbors represents the overall aggregation function that we aim to learn using our equivariant network. It maps the inputs to the output node embedding $n_i$'. Similarly, for edges, the updated node embeddings $e_{ji}'$ fed into a learnable function $\Phi_{ji}$ has the following form:

$$e_{ji}' = \Phi_{ji}(n_i', n_j', |r_{ji}|, \hat{r}_{ji}). \tag{2}$$

It maps the inputs consisting of the updated node embeddings to the edge embeddings. In our case, due to the large unfeasible size of the graph, we have to partition it into slices for training. In partitioned subgraphs, however, there are also connected neighbor nodes that lie outside of the partition, meaning that some of the $j$ terms in Eq. (1) are missing. Ignoring the contribution from these nodes leads to incomplete/wrong aggregation. As a result, the wrong aggregation function would be learned when fitting the final node output to the target Hamiltonian data (during minimization of MSE loss). See the ablation study in Section 4 and Table 3) for more details.

This is why we introduced virtual nodes and edges. They account for the presence of connected neighbors outside of the partition when computing the updated node embeddings for atoms situated within the partition:

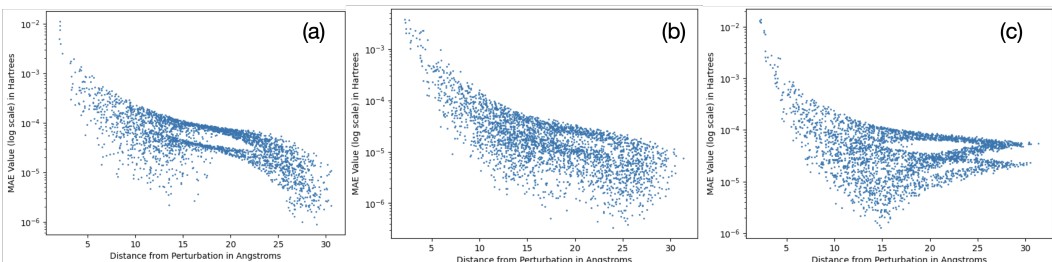

Figure 15: Scatter plots showing the decay of MAE with increasing distance from different perturbations, including (a) 0.1 Å translation of an O atom, (b) replacement of O atom with a vacancy, and (c) replacement of O atom with Hf atom

$$n'_{i\in\mathcal{P}} = \sum_{j\in\mathcal{N}(i)\cap j\in\mathcal{P}} \Phi_j(Z_i, Z_j, |r_{ji}|, \hat{r}_{ji}) + \sum_{j\in\mathcal{N}(i)\cap j\notin\mathcal{P}} \Phi_j(Z_i, Z_j, |r_{ji}|, \hat{r}_{ji}), \tag{3}$$

$$e'_{(i,j)\in\mathcal{P}} = \Phi_{ji}(n'_{i\in\mathcal{P}}, n'_{j\in\mathcal{P}}, |r_{ji}|, \hat{r}_{ji}), \tag{4}$$

where $\mathcal{P}$ is defined as the set of atoms belonging to the partition. The virtual nodes and edges are cast into the second summation on the right-hand-side of Eq. (3). They contain all necessary inputs ($Z_i$, $Z_j$, $|r_{ji}|$, $\hat{r}_{ji}$) needed to compute the aggregation function in the first layer of the network. Therefore, for the case of nodes within the partition ($n_{i\in\mathcal{P}}$), Eq. (3) is now equivalent to Eq. (1), as they contain the same terms and inputs. By extension, since the edge embeddings within the partition are only updated based on nodes within the partition (Eq. (4)), they are also correct, with Eq. (4) being equivalent to Eq. (2) for all $e_{(i,j)\in\mathcal{P}}$. Overall, in the case of a single MP layer, the local environment for nodes and edges within the partition is thus identical to that of the full graph. As a consequence, the correct aggregation function, along with accurate predictions, are obtained from training.

Note that throughout this process, the output embeddings of the virtual nodes and edges are not used at all and remain completely masked during training - only their inputs ($Z_i$, $Z_j$, $|r_{ji}|$, $\hat{r}_{ji}$) are used to inform the network. Their own local environment and outputs have, therefore, no influence on the aggregation function learned, and do not need to be corrected.

## K ANALYSIS OF PERTURBATION EFFECTS AT VARYING DISTANCES

To demonstrate the effects of long and short range perturbations in amorphous structures, we introduce a single perturbation at one chosen location in the structure and measure the mean absolute error of the onsite Hamiltonian blocks when compared to that of the unperturbed structure. The types of perturbations introduced include single oxygen atom translation, oxygen vacancy, and substitution of an oxygen by a hafnium atom, which are plotted against distance from perturbation in Fig. 15 (a), (b) and (c) respectively. In all cases, the effect of the perturbation rapidly decays with increasing distance.

For the case of a 0.1 Å translation perturbation, the average onsite MAE at a distance of 8 Å away is given by 0.15 $mE_h$. Considering the average value of an onsite Hamiltonian block (63 $mE_h$), the perturbation only affects the matrix elements by 0.24% overall. Similarly, for vacancy and substitution perturbations, the matrix elements of atoms located 8 Å away only changed by 0.18% and 0.12% respectively. This implies that for our chosen cutoff of 8 Å, perturbations occurring outside of the radius surrounding the atom have a negligible effect on its Hamiltonian matrix elements. This also means that the electronic structure of that atom can be learned using information from the local atomic environment.

This is also demonstrated through our study of sub-stoichiometric $HfO_x$ with randomly distributed vacancies. Despite training on independent slices, we ensure that every atom within that slice is

surrounded by a complete local environment with a radius of 8 Å through the use of virtual nodes and edges. Any vacancies outside of that radius have a negligible effect on the atom, and are seen and learned by other partitions. When multiple slices are trained together, the entire distribution of perturbations are captured, allowing the model to generalize well to unseen structures with a completely different distribution of vacancies and local atomic environments.

