# OpenReview forum: "Learning the Hamiltonian of Disordered Materials with Equivariant Graph Networks"
_ICLR.cc/2025/Conference — Submitted to ICLR 2025_

### Official Review · Reviewer_frUn · 2024-10-21

**Soundness:** 3
**Presentation:** 3
**Contribution:** 3
**Rating:** 6
**Confidence:** 3

**Summary:**

This paper presents an equivariant graph neural network (GNN) method based on EquiformerV2 to predict disordered materials’ ground state Hamiltonian matrix $\mathbf{H}$. An augmented partitioning approach is used to partition the large, highly connected graph, addressing the computational cost issue. The method is tested on an $\ce{H2O}$ benchmark and a custom dataset of large-scale amorphous $\ce{HfO2}$, showing accuracy and computational efficiency improvements. The effects of graph construction (cutoff radius and connectivity), model hyperparameters (virtual nodes, number of layers, etc.), and the augmented partitioning approach are investigated.

**Strengths:**

- This work tackles an important topic in materials science. Predicting the Hamiltonian to provide an accurate initial guess could speed up DFT calculations, thus improving the efficiency of computational materials modeling.
- The proposed partitioning approach leads to a considerable reduction in computational cost.

**Weaknesses:**

- The novelty and contributions should be better articulated. Currently, the Background and Related Works section mainly describes the physics. What past works this work is built upon and what are the new things should be made clear. Perhaps listing the contributions in the Introduction could be helpful.
- Some mathematical notations are unclear or confusing. (1) Scalars, vectors, and matrices should be distinguished using, e.g., boldface. (2) $N_A$ denoting the number of atoms could be confused with the Avogadro constant.

**Questions:**

- The proposed methods are mainly tested on the $\ce{HfO2}$ dataset, which does not convincingly show the generalizability to other disordered materials systems. Would the authors consider adding other test cases, or discussing the applicability and limitations?
- Sec 3.3 says “…generated a dataset of only three unique a-$\ce{HfO2}$ structures for training, validation, and testing”. How is “unique” defined? Could there be significant overlap (e.g., same local environments) between these three structures?
- How is the “near-sightedness” of $\mathbf{H}$ defined and supported? Does this hold strictly or approximately in theory?
- What metric is used to obtain the 0.87% accuracy, is it MAE? If so, is the mean value calculated over all eigenvalues? Given the large difference of magnitude between eigenvalues of $\mathbf{H}$, I’m not sure whether this is a reasonable metric.

---

> ### Author Response · Authors · 2024-11-20
> **Response to Reviewer frUn**
>
> **Notational clarity:** Thank you for pointing this out. We have gone through and standardized all the mathematical notation.
>
> **List of contributions**: Thank you for the suggestion - we have added a list of contributions to the **Introduction**. Please also see the ‘general response'.
>
>
> **Generalizability to other disordered systems:**
>
> We have now added two additional examples of structural disorder (to appendix I), as detailed in the ‘General response’ (substoichiometric HfOx and amorphous PtGe). Together, these examples demonstrate the capabilities of our approach in tackling different types of structural disorder.
>
>
> **Definition of unique:**
>
> We understand the concern - to help illustrate the uniqueness of the structures in the dataset, we have included a plot of the outliers in Hf coordination in **Fig. 9 in Appendix F**. Amorphous structures are not a completely random distribution of atoms, they typically contain some smaller structural motifs and a range of coordination numbers. However, the spatial distribution of these differently coordinated atoms is different in all samples, indicating that they are all unique.
>
>
> **Near-sightedness:**
>
> Yes, the near-sightedness principle holds strictly for electronic Hamiltonians represented in a local basis. Please see our general response for more details and the newly added **Fig. 5b**, which illustrates this effect for our example system.
>
>
> **Metric for accuracy:**
>
> For clarity, we changed the naming from relative MSE/MAE to relative L1 and L2 errors (see Fig. 6). In addition, we computed the same error metrics for the case where only the occupied states are considered (0.84%).  The metric is reasonable since the absolute errors shown in Fig. 13 in Appendix H are in the same order of magnitude at band edges.

---

> > ### Comment · Reviewer_frUn · 2024-11-22
> >
> > Thanks for addressing my questions. I will increase the Presentation and Soundness scores.

---

### Official Review · Reviewer_ZTdr · 2024-10-25

**Soundness:** 3
**Presentation:** 3
**Contribution:** 3
**Rating:** 6
**Confidence:** 5

**Summary:**

This paper presents an approach to learn the ground-state Hamiltonian of disordered materials using equivariant graph neural networks (GNNs). The key methodological contribution is an "augmented partitioning" scheme that aims to handle large atomic systems by slicing the graph into multiple partitions and adding virtual nodes/edges at boundaries. The authors demonstrate their method on amorphous HfO2 with 3,000 atoms, claiming acceptable accuracy compared to DFT calculations.

**Strengths:**

The main contribution lies in the augmented partitioning scheme that enables the application of equivariant GNNs to large-scale atomic systems. The work demonstrates systematic validation through comprehensive ablation studies and detailed analysis of computational performance, supported by comparison with DFT calculations on HfO$_2$ systems. The technical presentation is well-structured, with clear mathematical formulations and thorough documentation of the computational workflow. From a practical perspective, the work provides a solution for electronic structure prediction of large-scale systems (3000+ atoms), offering significant computational savings compared to traditional DFT calculations.

**Weaknesses:**

The paper has several fundamental methodological issues. The most critical concern is that the proposed partitioning scheme breaks the hierarchical nature of Message Passing Neural Networks. While standard MPNNs use multiple MP layers to incrementally increase the receptive field, this method's single-layer virtual node approach artificially truncates information flow at partition boundaries, compromising the network's ability to capture long-range interactions. The authors attempt to justify this limitation by invoking the "near-sightedness" of the Hamiltonian, but provide no rigorous analysis of how Hamiltonian matrix elements Hij are affected by distant atoms or how the neglected higher-order interactions might impact accuracy. The validation is severely limited, with only three structures of a perfect stoichiometric HfO2 being tested. In real materials applications, amorphous HfO2 often contains various defects such as oxygen vacancies and the Hf:O ratio can deviate from the ideal 1:2 stoichiometry. The absence of these realistic material features in the training data raises serious concerns about the method's practical utility. Without testing on structures with different stoichiometries, defects, and degrees of disorder, it's impossible to assess whether this approach can truly address real computational challenges in materials science.

**Questions:**

1.  Standard Message Passing Neural Networks (MPNNs) are designed to capture hierarchical structural information through multiple layers, where each additional MP layer extends the receptive field and allows nodes to aggregate information from more distant neighbors. The partitioning scheme presented in this paper fundamentally breaks this mechanism by using only single-layer MP with virtual nodes at boundaries. The authors shoule provided more  justification for using single-layer MP.  they should provide quantitative analysis demonstrating how Hamiltonian matrix elements Hij are affected by atoms at different distances, and how this relates to their architectural choices? This should include comparison with full multi-layer MP on smaller systems.

2. The work emphasizes parallel training through system partitioning to handle large disordered systems. However, if the locality of the Hamiltonian truly holds as claimed, then a model trained on small cells should be able to generalize to larger disordered systems, as the local atomic environments would be similar. In this case, standard GNN approaches without partitioning would remain valid.  The authors should provided quantitative comparison between their method  and a standard GNN model trained on small cells and directly applied to large systems .

3. The paper only presents the validation with perfect stoichiometric HfO$_2$ structures, which is far from realistic material conditions. Could the authors demonstrate how their method performs on structures wit Hf or O vacancies, varying Hf:O ratio.

4. The reported accuracy of 5.87 meV is significantly higher than the sub-meV accuracy achieved by state-of-the-art models in electronicHamiltonian prediction. The authors should either justify why this level of accuracy is sufficient, or provide direct comparisons with other methods (e.g., DeepH, HamGNN, etc.) on benchmark systems on small systems. If the accuracy loss is a trade-off for handling larger systems, this limitation should be explicitly discussed and quantified.

---

> ### Author Response · Authors · 2024-11-20
> **Response to Reviewer ZTdr (1/3)**
>
> **General**
>
> As was pointed out by another reviewer, we did not sufficiently stress the two contributions of our work (the development of a network for Hamiltonian prediction based on SO(2) convolutions by combining concepts from several previous works, and the augmented partitioning approach). We have now added a contributions section to the introduction.
>
> **(Q1) Nearsightendness principle + Analysis demonstrating how Hamiltonian matrix elements are affected by atoms at different distances**:
>
> We have added (1) an explanation of the near-sightedness principles of orbital interactions to **Section 3 (methods)**, and (2) a **sub-figure (Fig. 5 (b))** to illustrate the decay (plotted in log scale) of the Hamiltonian elements as a function of interatomic distance. When using a local basis, near-sightedness strictly holds and is a fundamental physical property of electronic Hamiltonians. **Previous studies on Hamiltonian predictions have similarly applied this principle to justify the omission of longer-range interactions [1]**.
>
> This information can also be interpreted from **Fig. 10** in **Appendix F.1**, where we plot the eigenvalue spectra of the ground-truth Hamiltonian after filtering the matrix elements by different values of $r_{cut}$. While an $r_{cut}$ value of $6Å$ results in a noticeable deviation of the eigenvalues from the unfiltered spectra, $r_{cut}$ = 8 and 10 $Å$ are identical to the unfiltered case. This is consistent with the results in **Table 2**, where the node and edge prediction errors saturate when determining the graph connectivity with $r_{cut}$ $\ge$ $8Å$.
>
> **(Q1) Justification of single message passing layer**
>
> There are two more points we would like to stress when it comes to determining the connectivity, and the use of a single layer. The cutoff radius is chosen to capture the relevant blocks of the Hamiltonian. This leads to the radius of connectivity being much larger compared to other networks, which represent materials as graphs and learn the properties of each node/atom, as these networks often define edges according to atomic bonds. With an $r_{cut}$ of $8Å$, the average atom has roughly ~180 edges to other atoms - the aggregation neighborhood is already large. Including a second-hop neighborhood would extend the receptive field to atoms that have blocks of negligibly small (often numerically zero) elements in the Hamiltonian, which do not provide any extra information for training, and also lead to neighborhood explosion considering the densely connected nature of these graphs, and the large cutoff, which would make training unfeasible. Finally, there are also examples of message-passing equivariant neural networks that have achieved success in the prediction of various complex properties of large scale systems while limiting the receptive field and enforcing locality (notably Allegro) [7].
>
> **(Q1) Use of smaller systems**:
>
> A crucial point that was not successfully conveyed in our initial submission is that it is impossible to generate small, non-crystalline unit cells for training. The reason why is illustrated in **Fig. 2** in the manuscript - **only large unit cells are able to capture realistic disorder**. To represent large unit cells that maintain the amorphous property of the material, large graphs are unavoidable. **However, within these large graphs, we can still exploit the locality of interatomic interactions to avoid computing long-range interactions**.
> If the application is for small molecules, s/p orbitals, or crystals where all information can be captured within a unit cell, then existing equivariant approaches such as DeepH2, DeepH-E3, QHNet, and HamGNN [2-5] will be able to treat them. Our network and training scheme are designed to operate in a regime that is inaccessible to these other works, and nevertheless presents significant scientific value and involves the highest computational burden to treat with conventional approaches. We have included an additional experiment in Table 3 of this response to show that our approach can similarly be applied to predict crystalline HfO₂ in the monoclinic phase, with a much better prediction accuracy compared to the amorphous HfO₂ structure.

---

> > ### Author Response · Authors · 2024-11-20
> > **Response to Reviewer ZTdr (2/3)**
> >
> > **(Q1/Q2) Comparison with other methods**
> >
> > We want to stress that methods such as DeepH, DeepH-E3, and HamGNN are unable to treat the problem we consider here, and can therefore not be used for direct comparison. Networks that rely on local rotations, such as DeepH, fail to treat the wide range of atomic motifs due to their use of a relative coordinate system for bond rotations, as is explained in the authors’ subsequent work [2]. HamGNN, DeepH-E3, and QHnet [3-5] use tensor products to mix features with different l while remaining rotationally equivariant - this makes it very compute- and memory-intensive to treat larger graphs, as explained in [2]. The SO(2) convolution approach that we use was designed to overcome the need for these expensive tensor products while maintaining SO(3)-equivariance, thus bringing training to manageable times for materials with high $l_{max}$, and enabling reasonable memory consumption for very large atomic graphs.
> >
> > **(Q4) Reported accuracy:**
> >
> > Nevertheless, we point out that HamGNN [5] achieves 1.5-3.5 meV accuracy on relatively small (<100 atoms) crystalline structures. Similarly,  a 2.3 meV accuracy was shown with DeepH2 [6], for structures with <= 150 atoms. Our final error for 3000 atoms can be further reduced from **5.87** to **3.55 meV** after training on two full structures (we have added this information to Appendix I.1). We have now also tested our model and approach on crystalline HfO₂ (Appendix I and Table 3 below). Due to the repeated nature of the periodic crystal, we can achieve sub-meV node prediction errors and near-sub-meV edge prediction errors
> >
> > The higher error for the amorphous structures is thus attributed to limited data availability (from the computational expense of generating new training structures) and dataset complexity. Regardless, we perform the eigenvalue spectra analysis in **Fig. 5** of the manuscript to demonstrate that this error is already sufficient for practical application.
> >
> > **(Q3) Other examples of structural disorder**:
> >
> > Atomic structures with different stoichiometry, crystallinity, and extended defects are part of our standard computational workflows. We have now added two additional examples of disorder (**substoichiometric HfOₓ, amorphous PtGe**):
> >
> > 1.**Sub-stoichiometric HfOₓ** (with 5%, 10%, 15% vacancies corresponding to x = 1.9, 1.8 and 1.7) was generated through the random insertion of oxygen vacancies into stoichiometric HfO₂ structures. More details on the generation of these structures can be found in Appendix I2. We have added a section detailing this procedure in Appendix I. Three models trained on a structure with 5%, 10% and 15% vacancy respectively were each tested on different unseen HfOₓ structures with vacancies (5%, 10% and 15%), with results shown in Table 1 below (more details on structures and training approach can be found in **Appendix I.2**). The $\epsilon_{n}$ and $\epsilon_{e}$ values across different experiments lie within a small range (2.44-2.94 $mE_h$ and 0.16-0.18 $mE_h$ respectively), showing that the network generalizes well to structures with different concentrations of vacancies, regardless of which vacancy configuration it is trained on.
> >
> > | **Training set** | **Testing set** | **εₙ [mEₕ]** | **σₙ [μEₕ]** | **εₑ [mEₕ]** | **σₑ [μEₕ]** |
> > |:-----------------:|:---------------:|:------------:|:------------:|:------------:|:------------:|
> > | 5%               | 5%             | 2.44         | 5.06         | 0.16         | 0.31         |
> > | 5%               | 10%             | 2.58         | 5.23         | 0.18         | 0.33         |
> > | 5%               | 15%             | 2.50         | 4.96         | 0.17         | 0.33         |
> > | 10%              | 5%            | 2.48         | 5.12         | 0.18         | 0.33         |
> > | 10%              | 10%            | 2.50         | 4.96         | 0.17         | 0.33         |
> > | 10%              | 15%            | 2.60         | 4.89         | 0.18         | 0.33         |
> > | 15%              | 5%             | 2.94         | 6.45         | 0.16         | 0.34         |
> > | 15%              | 10%             | 2.52         | 5.01         | 0.16         | 0.34         |
> > | 15%              | 15%             | 2.52         | 4.69         | 0.16         | 0.31         |
> >
> > Table 1: HfO₂ models trained and tested with different stoichiometry using augmented partitioning. 18 slices of each structure, each 3 Å thick, was used for training. The training method is identical to the one used to obtain Table 4. Three models (trained on structures with 5%,10 % and 15 % vacancy respectively) was tested on unseen test structures with vacancies ranging from 5% to 15%.

---

> > > ### Author Response · Authors · 2024-11-20
> > > **Response to Reviewer ZTdr (3/3)**
> > >
> > > To demonstrate that the augmented partitioning approach similarly does not affect accuracy for sub-stoichiometric HfO₂, full graph training was also compared with the augmented partitioning training approach in Table 2. Both models were trained using the same structure with **15%** vacancy (with and without partitioning). From the minimal difference in ϵn and ϵe values between full and partitioned approaches, it can be seen that both approaches generalize equally well to different vacancies. These values are also close to that of stoichiometric HfO₂ in Table 4 in the paper,therefore showing that the augmented partitioning approach can also be applied even in the case of more realistic sub-stoichiometric structures.
> > >
> > >
> > >
> > > | **Training method** | **Oxygen vacancies** | **εₙ [mEₕ]** | **σₙ [μEₕ]** | **εₑ [mEₕ]** | **σₑ [μEₕ]** |
> > > |:-------------------:|:--------------------:|:------------:|:------------:|:------------:|:------------:|
> > > | partitioned         | 5%                  | 2.94         | 6.45         | 0.16         | 0.34         |
> > > | partitioned         | 10%                  | 2.52         | 5.01         | 0.16         | 0.34         |
> > > | partitioned         | 15%                  | 2.52         | 4.69         | 0.16         | 0.31         |
> > > | full                | 5%                  | 2.96         | 6.07         | 0.19         | 0.38         |
> > > | full                | 10%                  | 2.67         | 5.18         | 0.18         | 0.36         |
> > > | full                | 15%                  | 2.64         | 4.83         | 0.17         | 0.35         |
> > >
> > >
> > >
> > > Table 2: Comparison between full graph training and the augmented partitioning training (18 slices with 3 Å thickness) using the same HfO₂ structure with 15% vacancies. Models are tested on structures with vacancies ranging from 5% to 15%.
> > >
> > >
> > > 2.**Amorphous Pt-doped Germanium**. We tested our augmented partitioning approach on two amorphous PtGe structures (structure 1 and 2), each containing 2688 atoms (Pt:Ge ratio is 1:2). The trained model was tested on a full unseen structure (structure 2), with results shown in Table 3. This is also an example of doped structure where the insertion of Pt breaks the stoichiometry of the Ge structure. **The final obtained error for all 2688 atoms and 2148055 edges is 1.43 meV**, much lower than that of HfO₂, even when trained on only a single slice. This therefore demonstrates the generalizability and robustness of the augmented partitioning approach when applied to completely different dataset with a much larger cutoff radius.
> > >
> > >
> > >
> > > | **Material**         | **Cutoff [Å]** | **εₙ [mEₕ]** | **σₙ [μEₕ]** | **εₑ [mEₕ]** | **σₑ [μEₕ]** | **εₜₒₜ [mEₕ]** | **εₜₒₜ [meV]** |
> > > |:---------------------:|:--------------:|:-------------:|:-------------:|:-------------:|:-------------:|:---------------:|:--------------:|
> > > | Crystalline HfO₂      | 8              | 0.01          | 0.02          | 0.04          | 0.07          | 0.04            | 1.17           |
> > > | Amorphous HfO₂        | 8              | 2.29          | 5.09          | 0.20          | 0.36          | 0.22            | 5.87           |
> > > | Amorphous PtGe        | 16             | 0.87          | 1.43          | 0.05          | 0.10          | 0.05            | 1.43           |
> > >
> > >
> > >
> > > Table 3: Summary of models trained on crystalline HfO₂, amorphous HfO₂, and amorphous PtGe materials, respectively. The HfO₂ model was trained on different slices of the same crystalline structures. On the other hand, the PtGe model was trained on one slice of structure 1 and tested on a fully unseen structure 2.
> > >
> > > Together, these examples demonstrate the capabilities of our approach in tackling different types of structural disorder. The reason why our approach generalizes to different materials comes from the fact that we do not sample from the materials design space, but from the orbital one. While different materials contain atoms with different atomic numbers, which act as identifiers, they crucially all use the same set of local basis functions. If the network works on a material characterized by s,p,d orbitals, then we expect it to work with any other material whose description relies on the same set of orbitals, the inputs being orbitally- and not atomically-resolved.

---

> > > > ### Author Response · Authors · 2024-11-20
> > > > **Response to Reviewer ZTdr (References)**
> > > >
> > > > **References**
> > > >
> > > > [1] Li, H., Wang, Z., Zou, N. et al. Deep-learning density functional theory Hamiltonian for efficient ab initio electronic-structure calculation. Nat Comput Sci 2, 367–377 (2022).
> > > >
> > > > [2] Yuxiang Wang, He Li, Zechen Tang, Honggeng Tao, Yanzhen Wang, Zilong Yuan, Zezhou Chen, Wenhui Duan, and Yong Xu. Deeph-2: Enhancing deep-learning electronic structure via an equivariant local-coordinate transformer, 2024a
> > > >
> > > > [3] Gong, X., Li, H., Zou, N. et al. General framework for E(3)-equivariant neural network representation of density functional theory Hamiltonian. Nat Commun 14, 2848 (2023).
> > > >
> > > > [4] Haiyang Yu, Zhao Xu, Xiaofeng Qian, Xiaoning Qian, and Shuiwang Ji. Efficient and equivariant graph networks for predicting quantum hamiltonian, 2023b.
> > > >
> > > > [5] Zhong, Y., Yu, H., Su, M. et al. Transferable equivariant graph neural networks for the Hamiltonians of molecules and solids. npj Comput Mater 9, 182 (2023).
> > > >
> > > > [6] Yuxiang Wang, Yang Li, Zechen Tang, He Li, Zilong Yuan, Honggeng Tao, Nianlong Zou, Ting Bao, Xinghao Liang, Zezhou Chen, Shanghua Xu, Ce Bian, Zhiming Xu, Chong Wang, Chen Si, Wenhui Duan, Yong Xu, Universal materials model of deep-learning density functional theory Hamiltonian, Science Bulletin, Volume 69, Issue 16, Pages 2514-2521 (2024).
> > > >
> > > > [7] Musaelian, A., Batzner, S., Johansson, A. et al. Learning local equivariant representations for large-scale atomistic dynamics. Nat Commun 14, 579 (2023).

---

> > ### Comment · Reviewer_ZTdr · 2024-11-25
> > **Further questions about the locality**
> >
> > The authors' analysis of Hamiltonian locality is incomplete and potentially misleading. Let me explain my concerns in detail.
> > While the authors demonstrated that Hij decays with increasing interatomic distance rij, this only establishes the sparsity pattern of the Hamiltonian matrix - a well-known property. However, they failed to address the more fundamental question of locality: whether the significant Hamiltonian matrix elements (those between nearby atoms i and j) can be accurately determined using only local atomic environments.
> > To properly establish the locality approximation, the authors need to demonstrate that Hij can be accurately predicted using only local structural information. Specifically, for atom pairs (i,j) where rij is small and Hij is significant, they should:
> >
> > 1. Systematically perturb the positions of distant atoms k through various mechanisms:
> >     - Atomic displacements
> >     - Introduction of defects
> >     - Atomic substitutions
> > 2. Quantify the response of Hij to these perturbations
> > 3. Demonstrate that this response decays rapidly with increasing distances |rk-ri| and |rk-rj|
> >
> > Simply showing that Hij becomes small for large rij is insufficient - this merely establishes matrix sparsity. The crucial physical question is whether the non-zero elements of the Hamiltonian can be determined from local structural information alone. This distinction is particularly important for amorphous systems, where local structural disorder could potentially lead to more complex non-local effects.

---

> > ### Comment · Reviewer_ZTdr · 2024-11-25
> > **Further concerns about the  single message passing layer**
> >
> > The authors' justification for single-layer message passing reveals a critical physical limitation. Single-layer MP can only capture pairwise (2-body) interactions between atoms, effectively modeling Hij as a sum of independent 2-body terms with neighboring atoms. However, in reality, Hij should depend on the collective electronic environment through many-body effects, which are particularly important in disordered systems. Multiple MP layers are necessary to capture these higher-order correlations.
> > Therefore, the authors' argument that single-layer MP is sufficient because "Hij decays with distance" misses the fundamental physics: while Hij may decay spatially, its value for nearby atoms depends on complex many-body correlations that cannot be captured by pairwise interactions alone. This limitation could significantly impact the model's accuracy, especially for disordered systems.

---

> > ### Comment · Reviewer_ZTdr · 2024-11-25
> > **further questions about using smaller systems:**
> >
> > The authors' argument about system size requirements needs clarification. While it's true that large unit cells are necessary to properly describe amorphous structures, this doesn't necessarily require training on entire large systems. If the claimed locality of Hamiltonian matrix elements holds, then:
> >
> > One could train the model using local environments extracted from large systems. These local environments would contain all the necessary structural information to predict Hij. The trained model should then be transferable to both small and large systems
> > This approach would be more computationally efficient while still capturing the essential physics of the amorphous system. The authors should either implement such a local environment training scheme or explain why it wouldn't work in their case.

---

> ### Author Response · Authors · 2024-11-26
> **Response to Reviewer ZTdr (Using smaller systems)**
>
> In the reply. it was mentioned that: "One could train the model using local environments extracted from large systems. These local environments would contain all the necessary structural information to predict Hij. The trained model should then be transferable to both small and large systems."
>
> Training using local environments extracted from large systems is far from trivial. **This is, however, exactly what our augmented partitioning approach was designed for**. Let us explain in further detail what the challenges are and why our method addresses them:
>
> To extract a local environment from large systems and use it for training, three relatively “straightforward” naive approaches can be envisioned:
>
> 1. Extracting small samples from large amorphous structures, and then performing DFT calculations on these isolated small unit cells. The DFT results from different samples are then used to train the model and predict the large amorphous structures.
>
> 2. Same method as 1, but with the addition of periodic boundary conditions to the small samples to account for the influence of the atomic environment.
>
> 3. Performing DFT on a full graph and using naive partitioning to extract a local sub-graph from the full graph for training, ignoring connections to other slices.
>
> **Approach 1:**
> The main issue with the first approach comes from the boundary conditions that should be applied during the DFT calculations. The small samples we can extract from the large amorphous structure are densely connected to other atoms. Neglecting the latter and considering only the isolated parts, which are treated as molecules, **gives rise to numerous dangling bonds at the surface of the small samples and highly inaccurate Hamiltonian matrix elements**.
>
> To further substantiate this point, we performed additional DFT calculations on an isolated sample of size 10x10x10 Å extracted from the large amorphous HfO₂ structure, placed in vacuum, i.e., without periodic boundary conditions. We then compared the resulting Hamiltonian matrix elements to those of the same atoms, but taken from the Hamiltonian matrix corresponding to the original large structure. The Mean Absolute Error (MAE) for the onsite elements of the small sample is **752 meV**. In other words, the Hamiltonian matrix elements of the small sample are **completely different from their counterparts** in the large structure. Hence, cannot be used for training the model at all. Furthermore, **the offsite Hamiltonian blocks representing the connections between the isolated piece and the rest of the large amorphous structure will also be missing from the training data**.
>
> **Approach 2:**
> As a second approach, periodic boundary conditions could be added to the small samples to avoid dangling bonds and mimic the atomic environment. However, for small unit cells, periodic boundary conditions typically lead to strong interactions between one atom and its periodic replica and thus very different Hamiltonian matrix entries than those of the original, large structure. We repeated the previous experiment with the inclusion of periodic boundary conditions: the onsite MAE becomes even larger (**813 meV**). The inaccuracies caused by the consideration of small samples only are the reason (i) why DFT calculations of large disordered structures is imperatively needed despite being computationally expensive, and (ii) why the learning/prediction of such systems has so far remained an open issue in literature.
>
> **Approach 3:**
> Alternatively, as a third approach, DFT calculations can be performed on a full graph and, through “naïve" partitioning, a local sub-graph can be extracted from the full graph for training, i.e., all connections between the sub-graph and rest of the graph are ignored. As mentioned in the manuscript, the omission of connections to atoms outside of the partition leads to the wrong/incomplete aggregation of information. To learn the correct aggregation function and be able to make accurate predictions, we must preserve the full connectivity of the partition subgraph. The poor prediction accuracy of this this third approach can be found in the **ablation study (Table 3 of the manuscript)**.
>
> **This brings us to our proposed augmented partitioning approach, which addresses this connectivity issue by introducing virtual nodes and edges**. Through this method, the local environment is correctly extracted from the large graph, which also contains which also contains accurate data from the DFT computation of the full amorphous structure. As a result, the prediction results using this approach is on par with other literature (refer to previous response on comparison with literature).
> We hope that we have now addressed this question. Please let us know if there is anything else we missed.
>
> **Note that the responses to the other two questions (single message passing layer and perturbation) will also be posted soon.**

---

> > ### Comment · Reviewer_ZTdr · 2024-11-26
> >
> > Thank you for your detailed explanation. I understand your points regarding approaches 1 and 3. However, I'm still a little confused about your analysis of approach 2 (using periodic boundary conditions on small samples). In the DeepH papers [H. Li, et al., Nat Comput Sci 2, 367 (2022) and X. Gong, et al., Nat Commun 14, 2848 (2023)], the authors demonstrate an experiment with twisted bilayer graphene. Although this system requires large cells due to moiré patterns, they successfully trained their model using non-twisted small cells and then generalized it to the large twisted structure. This seems to contradict your assertion about the ineffectiveness of training on smaller periodic systems. Could you explain why their approach works in this case, despite the challenges you've outlined for periodic boundary conditions on small samples?
> >
> > Thank you again for addressing my concerns and I look forward to your responses regarding the single message passing layer and perturbation questions.

---

> ### Author Response · Authors · 2024-11-27
> **Response to Reviewer ZTdr (small systems, twisted bilayer graphene)**
>
> The key difference between the systems we are investigating and twisted bilayer graphene, is the **absence of atomic order/periodicity in our amorphous structures, while periodicity is still present in twisted bilayer graphene**.
>
> While it is true that twisted bilayer graphene with varying twist angles relies on large unit cells to be described, **periodicity still exists in these structures along the two directions in the graphene plane**. The main additional information that must be learned in this case is the interaction between the different graphene layers in the third direction, which can be captured by the small samples. The local atomic environments of the atoms within the large bilayer structure are therefore still highly similar to that of the small training samples with periodic boundary conditions applied in the graphene plane. As a result, a model trained on information from small unit cells can generalize to test structures that are larger along these periodic directions.
>
> In our case, **there is no repeated motif/structure along any direction because of the amorphous nature of our samples.** Applying periodic boundary conditions to a small unit cell would lead to a crystalline material that does not resemble our large amorphous structures at all. It is essential to note that amorphous HfO2 contains a large variety of different motifs and local environments that, when connected together, exhibit the unique properties that we are trying to predict. Replacing the disordered environments and connections with periodic images leads to the loss of amorphous behavior, and very different Hamiltonian elements. We hope this fully addresses your question.

---

> ### Author Response · Authors · 2024-11-27
> **Response regarding perturbations**
>
> To demonstrate the effect of long and short range perturbations in amorphous structures, we introduced a single perturbation at one chosen location in the sample and measured the mean absolute error of the resulting onsite blocks of the Hamiltonian matrix when compared to that of the unperturbed structure.
>
> The types of perturbation introduced include  single atom translation (0.1 Å shift), vacancy (replacing O atom), and substitution (O atom replaced with Hf atom). Their influence is plotted against distance from perturbation in **Fig. 15 (a), (b) and (c)** respectively (**In Appendix K of the updated manuscript**). In all cases, the effect of the perturbation rapidly decays with increasing distance.
>
> For the case of a 0.1 Å translation perturbation,  the average onsite MAE at a distance of 8 Å away is equal to $0.15$  $mE_h$. Considering the average value of an onsite Hamiltonian block ($63$ $mE_h$), the perturbation only affects the matrix elements by **0.24\%** overall. Similarly, for vacancy and substitution perturbations, the matrix elements of atoms located 8 Å away only changes by **0.18\%** and **0.12\%**, respectively. This implies that for our chosen cutoff of 8 Å, perturbations occurring outside of the selected range around one specific atom have a negligible impact on its Hamiltonian matrix elements. This also means that the electronic structure of that atom can be learned using information from the local atomic environment only.
>
> This feature is also demonstrated through our study of sub-stoichiometric HfO$_x$ with randomly distributed vacancies. Despite training on independent slices, we ensure that every atom within that slice is surrounded by a complete local environment, with the inclusion of all neighbors within a radius of 8 Å. Any vacancies outside of that range have a negligible effect on the atom, and are seen and learned by other partitions.  When multiple slices are trained together, the entire distribution of perturbations are captured, allowing the model to generalize well to unseen structures with a completely different distribution of vacancies and local atomic environments.

---

> ### Author Response · Authors · 2024-11-29
> **Response regarding single message passing layer limitation**
>
> First, we would like to emphasize that the augmented partitioning approach we propose is a **general training method that can be applied to other, more expressive networks**. It is not strictly attached to the EquiformerV2/DeepH2 architecture that we implemented in this work. The augmented partitioning approach is mainly applied during graph construction, which is mostly independent of the architecture itself.
>
> Second, we want to point out that similar problems have been successfully treated by networks which only use strictly local interactions. The best example of this is Allegro [1], which has been used to predict energy and forces, and build interatomic potentials. It takes in a similar set of two-body inputs (atomic numbers of atom i and neighbors j, and their distances) as our architecture, requiring no information from atoms beyond the first hop neighborhood. **Many body representations can then be constructed through repeated tensor products as more layers are added.** It can also be adapted for the prediction of Hamiltonian entries as done by this recent ICLR submission: https://openreview.net/forum?id=kpq3IIjUD3. This is consistent with how local information is sufficient to achieve the prediction accuracies we have shown.
>
> We can combine this approach with our augmented partitioning method to capture higher body order information through more layers and further improve our prediction accuracy on amorphous materials (**which is already comparable to literature**), while maintaining the same receptive field. This is a promising direction to explore that is beyond the scope of the paper.
>
> What we have demonstrated in this particular submission using the current architecture, is that the proposed augmented partitioning approach gives a similar level of prediction accuracy compared to full graph training when the receptive field is restricted. We have therefore developed a **generally applicable partitioning method that enables the reliable training and prediction of large amorphous structures that were so far not achievable.**
>
> References:
> [1] Musaelian, A., Batzner, S., Johansson, A. et al. Learning local equivariant representations for large-scale atomistic dynamics. Nat Commun 14, 579 (2023).

---

> > ### Comment · Reviewer_ZTdr · 2024-12-02
> >
> > Thank you for addressing my questions. I agree that combining the local updating message-passing approach and partitioning parallelization presents a promising direction. In light of this, I will increase the score.

---

### Official Review · Reviewer_cPXL · 2024-11-03

**Soundness:** 3
**Presentation:** 2
**Contribution:** 2
**Rating:** 6
**Confidence:** 3

**Summary:**

The paper aims to develop a deep learning model, specifically leveraging Graph Neural Networks (GNNs), that can predict the ground-state Hamiltonian matrix (a representation of electronic properties) of disordered materials, which efficiently predicting the electronic properties of disordered materials compared to the traditional methods, such as Density Functional Theory (DFT). What’s more, this work lies in adapting equivariant GNNs and introducing an augmented partitioning approach to efficiently scale the model for large disordered systems.

**Strengths:**

1.The paper applies the equivariant GNNs combined with augmented partitioning is a significant methodological advancement. It allows the model to scale to large disordered systems, which is a critical capability in material science applications.
2.The authors provide an extensive suite of experiments that test the model’s accuracy, scalability, and generalization ability. The ablation studies in particular strengthen the claim that virtual nodes are essential for capturing disordered materials' electronic properties.
3.This work holds potential for practical applications, as it reduces the dependence on computationally intensive DFT simulations. The approach could pave the way for faster exploration of disordered materials in fields like electronics and energy storage.

**Weaknesses:**

1.The paper provides empirical discussions regarding the connectivity and choice of cutoff radius, it does not delve deeply into the theoretical basis behind these choices.
2. The paper lacks comprehensive experiments to illustrate the specific capabilities and limitations of the proposed GNN architecture in predicting Hamiltonian matrix elements across a range of disordered materials. Additional benchmarks comparing the model’s performance on Hamiltonian matrix predictions with different materials or conditions would strengthen the empirical foundation and highlight the robustness of the GNN approach for diverse applications.

**Questions:**

1. Can additional experiments be conducted on other datasets? For instance, it would be beneficial to see the model tested on larger graph datasets that necessitate a higher cutoff radius. Are there any practical limitations to testing on much larger systems that the authors should discuss?

---

> ### Author Response · Authors · 2024-11-20
> **Response to Reviewer cPXL**
>
> **Graph connectivity/cutoff radius:**
>
> The cutoff radius (which determines the connectivity) is a convergence parameter that can be chosen either by looking at the changes in the eigenvalue spectra (as in Fig. 10, where $r_{cut}$ was chosen such that the eigenvalue accuracy is not significantly compromised) or by checking the prediction accuracy of models with different $r_{cut}$ (as in Table 2) and choosing an $r_{cut}$ that achieves sufficient accuracy. However, it is not possible to analytically compute this quantity ahead of the DFT calculations, using only the structural (and basis) information. **A higher $r_{cut}$ implies that more interactions are captured (as shown for both HfO2 and PtGe in Table 1 below). Therefore, the general rule for an unknown material is to choose the maximum possible cutoff such that the slice fits into memory, in order to maximize prediction accuracy**.
>
> **Range of disordered materials:**
>
> Please see our general response, which details the changes we made to address this point (as it was brought up by multiple reviewers). In short, we agree that showing only one material is not a convincing demonstration of our application’s ability to handle electronic properties in the case of structural disorder. To fix this, we have added examples of **sub-stoichiometric HfOₓ (containing vacancies) and amorphous PtGe**. We are exploring the latter for its phase-change-related properties. The results have been added to **Appendix I** of the revised manuscript.
>
> To add to the response to the previous question on cutoff radius, we conduct a similar investigation on a completely new **amorphous PtGe** material, with results shown in Table 1 (also added to Appendix I.3). Here, as the PtGe structure occupies less memory, we can choose a very large cutoff radius of 16 Å without exceeding memory limits (unlike HfO2), and can therefore achieve a much higher prediction accuracy of **1.43 meV for all 2688 atoms and 2148055 edges** using only one slice for training. The results from Table 3 below (and Table 2 in the manuscript) also show that higher $r_{cut}$ is always preferred over lower $r_{cut}$ due to the significant improvements in overall accuracy. More details on the PtGe structure can be found in **Appendix I**.
>
>
> | **Cutoff [Å]** | **εₙ [mEₕ]** | **σₙ [μEₕ]** | **εₑ [mEₕ]** | **σₑ [μEₕ]** | **εₜₒₜ [mEₕ]** | **εₜₒₜ [meV]** |
> |:--------------:|:------------:|:------------:|:------------:|:------------:|:--------------:|:--------------:|
> | 6              | 0.87         | 1.43         | 0.15         | 0.19         | 0.17           | 4.60           |
> | 8              | 0.87         | 1.42         | 0.09         | 0.16         | 0.10           | 2.73           |
> | 16             | 0.87         | 1.43         | 0.05         | 0.10         | 0.05           | 1.43           |
>
> Table 1: Prediction accuracy of the model on amorphous PtGe with different $r_{cut}$
>
>
> We hope these additional tests and the high prediction accuracy validate the robustness of our approach, especially when applied to new datasets with larger cutoff radius.
>
>
> **Practical limitations to testing larger systems:**
>
> The idea behind this approach is that the training structures of a given size should sufficiently sample the range of atomic motifs which can be found in the structure, such that testing/prediction can be done on larger scales at similar prediction accuracy. We therefore do not expect any hard limitations in extending the augmented partitioning method to test on larger systems, which would require more augmented partitions. However in terms of training, there may be a limitation in terms of scaling due to the two issues (virtual node overhead + load imbalance) described in **Appendix G (please see Fig. 11)**. When training larger graphs, these imbalances could affect performance at larger scales.

---

### Official Review · Reviewer_4ZDn · 2024-11-04

**Soundness:** 3
**Presentation:** 3
**Contribution:** 3
**Rating:** 5
**Confidence:** 2

**Summary:**

The paper presents an efficient adaptation of equivariant graph neural networks for learning the electronic properties of disordered materials. It introduces an 'augmented partitioning' approach that divides large structures into smaller, manageable graphs, each enhanced with masked virtual nodes and edges while maintaining correct atomic neighborhoods during message passing. This method enables the effective learning of electronic properties, providing a scalable alternative to computationally intensive density functional theory (DFT) calculations.

**Strengths:**

The paper extends the application of equivariant neural networks to a challenging scientific problem that traditionally requires significant computational resources. It offers a specific strategy to address the computational cost and enhance the scalability of the implemented equivariant neural networks.

**Weaknesses:**

- While the proposed methods are promising for the specific task addressed, they appear to be limited to this particular problem domain. I suggest the authors apply their proposed graph partitioning methods to other materials-related challenges beyond the scope of this paper. For example, exploring applications in molecular modeling, crystalline structures, or other materials science problems would significantly broaden the impact of the work. This could be complemented by additional practical evaluations, such as comparing the method’s performance on various datasets or against other state-of-the-art approaches. Including metrics such as computational efficiency and accuracy in these broader contexts would provide a stronger empirical basis for the proposed method.

- I think this paper is not theoretically solid enough to meet the standards expected at this conference. The contributions are modest and mainly focused on a graph partitioning approach, which lacks a strong theoretical foundation. To enhance the work, I recommend the authors further develop the theoretical framework, particularly by providing more rigorous analysis or proofs regarding the scalability or performance guarantees of their approach.

**Questions:**

- The subject appears to be using an outdated template (ICLR 2024).
- Please address my concerns in the weakness section.

---

> ### Author Response · Authors · 2024-11-20
> **Response to Reviewer 4ZDn (1/2)**
>
> We realize that we did not sufficiently detail the novelty of our contribution in the initial submission. We added the following  ‘contributions’ section to the **Introduction**:
>
> 1. We develop an efficient GNN-based model for electronic property prediction by combining (1) the SO(2) convolution approach detailed in [1], (2) the equivariant attention mechanism proposed in [2], and (3) concepts from [3] and [4] to introduce learnable node/edge embeddings and a basis transformation layer for mapping outputs to the Hamiltonian. We provide the code for this implementation [in the Supporting Materials].
>
> 2. As the graphs required to capture amorphous disorder are necessarily large and incur high memory consumption in a full-batch training environment, we propose an efficient augmented partitioning method that partitions the input graphs and corrects atomic environments with masked virtual nodes and edges. Our method allows for arbitrarily large graphs to be broken down into partitions that can fit into GPU memory during training without compromising the achievable testing accuracy.
>
>
> **Problem specificity:**
>
> We agree that our methodology is specifically designed for electronic property prediction, from the construction of equivariant networks for Hamiltonian blocks to the augmented partitioning approach designed to tackle full-batch training of large, densely-connected graphs. Both of these techniques can find other applications that will be explored in future work.
>
> However, we do not want to stray from our application focus in this paper because this problem is of high scientific relevance. Predicting the electronic properties of large disordered structures from first-principles using density functional theory is a key research activity in materials science. It consumes a large portion of the computing time of several supercomputers in the world [5]. We have tried to stress the potential scientific impact of research in this area in the **Outlook** section of the manuscript. More generally, several previous papers at top venues have been dedicated specifically to predicting electronically-resolved information, several of which are cited in our work [6,7]. We strongly believe that this application is of significant interest to ICLR.
>
> Our approach is similarly straightforward when extending to crystalline materials. This is because crystalline materials are composed of a periodic tiling of a unit cell in space, with atomic environments varying only within the unit cell. When represented on a local basis, the corresponding Hamiltonian matrix is composed of identical sub-blocks. This also implies that the difference between training and testing data is almost negligible since the same/very similar local atomic environments are seen in both cases. After developing a similarly-sized structure of monoclinic HfO₂ (a crystalline phase), for example, we can achieve near sub-meV accuracy (more information can be found in **Appendix I**).  **On the other hand, amorphous structures contain a large range of different local atomic configurations that vary across different samples, necessitating the use of large unit cells to be fully captured. The accurate prediction of electronic properties of amorphous materials therefore remains an open problem in literature [8].** The approach introduced in our paper tackles this challenging problem.
>
> It is not straightforward to compare to our work to other studies, as no existing equivariant network that we are aware of is computationally capable of treating our datasets without extensive additional computational optimization. However, we can show generalizability to different disordered systems. In **Appendix I**, we have added examples of sub-stoichiometry HfOₓ ( with $x < 2$ ) and amorphous Pt-doped Ge, as mentioned in the ‘general response’ above. The code we provided in the Supporting Information also includes several examples applied to small-molecule datasets made from MD17 trajectories, including H₂O, uracil, and malondialdehyde. Results for one example - H₂O - are shown in **Appendix B** of our paper.

---

> > ### Author Response · Authors · 2024-11-20
> > **Response to Reviewer 4ZDn (2/2)**
> >
> > **Theoretical framework:**
> >
> > We are aware that it should be unnecessary to read the appendix in order to understand the main text. However, this is where we placed a lot of methodological details and more extensive experiments, which deviate from the flow of the paper. For example, we provide scaling studies of the partitioning approach in **Appendix G.2** together with the measured memory consumption. Since the partitions can be treated as separate batches, scaling is limited only by load imbalance and the small overhead introduced by edges from virtual nodes.
> >
> > Regardless, we understand that we likely did not provide enough detail. At the same time, we want to avoid repeating concepts that have been developed and explained in the cited works, such as the reduction of SO(3) to SO(2) operations [1]. If the reviewer is willing to clarify, we would appreciate knowing which parts of the framework can be further theoretically developed in order to reach a higher standard.
> >
> > **Template:**
> >
> > Thank you for pointing this out. We mistakenly used the old template. This has been fixed in the updated pdf.
> >
> >
> > **References**
> >
> > [1] Passaro, S. &; Zitnick, C.L.. (2023). Reducing SO(3) Convolutions to SO(2) for Efficient Equivariant GNNs. <i>Proceedings of the 40th International Conference on Machine Learning</i>, in <i>Proceedings of Machine Learning Research</i> 202:27420-27438
> >
> > [2] Yi-Lun Liao, Brandon Wood, Abhishek Das, and Tess Smidt. Equiformerv2: Improved equivariant transformer for scaling to higher-degree representations, 2023.
> >
> > [3] Gong, X., Li, H., Zou, N. et al. General framework for E(3)-equivariant neural network representation of density functional theory Hamiltonian. Nat Commun 14, 2848 (2023).
> >
> > [4] Yuxiang Wang, He Li, Zechen Tang, Honggeng Tao, Yanzhen Wang, Zilong Yuan, Zezhou Chen, Wenhui Duan, and Yong Xu. Deeph-2: Enhancing deep-learning electronic structure via an equivariant local-coordinate transformer, 2024a
> >
> > [5] Chang, C., Deringer, V.L., Katti, K.S. et al. Simulations in the era of exascale computing. Nat Rev Mater 8, 309–313 (2023).
> >
> > [6] Haiyang Yu, Zhao Xu, Xiaofeng Qian, Xiaoning Qian, and Shuiwang Ji. Efficient and equivariant graph networks for predicting quantum hamiltonian, 2023b.
> >
> > [7] Li, H., Wang, Z., Zou, N. et al. Deep-learning density functional theory Hamiltonian for efficient ab initio electronic-structure calculation. Nat Comput Sci 2, 367–377 (2022).
> >
> > [8] Reiser, P., Neubert, M., Eberhard, A. et al. Graph neural networks for materials science and chemistry. Commun Mater 3, 93 (2022).

---

> > > ### Comment · Reviewer_4ZDn · 2024-11-25
> > > **Response to Authors**
> > >
> > > I appreciate the authors' efforts in clarifying their contributions and providing further explanation. Overall, I find this is an interesting paper that addresses scientifically challenging problems. While the proposed networks heavily rely on existing literature, the introduced graph representation for large materials appears simple yet novel.
> > >
> > > However, I would like to inquire whether the authors can provide any theoretical support for their proposed graph methods. Specifically, how do the slicing, virtual nodes, and edge methods ensure accurate predictions of the desired properties? If the authors can theoretically substantiate this aspect, I would be inclined to adjust my rating to at least borderline accept.

---

> ### Author Response · Authors · 2024-11-26
> **Response to Reviewer 4ZDn (Theoretical Justification)**
>
> We are glad to have addressed some of your concerns. In response to your question, we have recently added the theoretical justification of the augmented partitioning approach to **Appendix J** of the manuscript. There, we added Equations 1-4 and explained them in detail. Hope it addresses your question. Please let us know if there are further clarifications needed. (e.g. notations etc.)

---

> > ### Comment · Reviewer_4ZDn · 2024-11-26
> > **Further Response to Authors**
> >
> > I appreciate the authors' further response to my comments. However, I find the newly added theoretical justification to be vague and difficult to follow. Upon reviewing this section, it seems to primarily reformulate the central methods rather than provide a rigorous theoretical analysis. Furthermore, there is even no clear theoretical statement or proposition that the authors aim to prove, which diminishes the clarity and significance of this addition. I strongly recommend that the authors refer to classic papers on the expressiveness of graph-based methods and attempt to formulate at least one concrete statement or hypothesis that they believe to be true. This would greatly enhance the theoretical grounding of the paper.

---

> ### Author Response · Authors · 2024-11-29
> **Response to Reviewer 4ZDn (further question on theoretical justification)**
>
> Thank you for the constructive feedback. Appendix J.1 is indeed a reformulation of the update rule behind augmented partitioning. When prompted to provide a more rigorous mathematical analysis, specifically in ‘how the augmented partitioning approach guarantees accurate node and edge predictions’, we concluded that the best approach would be to show that ‘the individual feature update rule is unchanged by the augmented partitioning approach’. Each embedding thus undergoes a standard message passing procedure. The hypothesis statement in this case, is that for node and edges within the partition, the feature update using augmented partitioning is equivalent to the feature update for full graph training for the first layer.
>
> We would also like to clarify that this paper is submitted to the applications track, which in our experience typically involves adapting ML architectural concepts for downstream scientific research use. During this process, we do not make contributions to mathematical foundations of machine learning, or specifically to the expressiveness of GNNs.  We follow the example of previously accepted ICLR papers in the area of materials science and chemistry that have also proposed new approaches (https://openreview.net/forum?id=CUfSCwcgqm, https://openreview.net/forum?id=KwmPfARgOTD, https://openreview.net/forum?id=mCOBKZmrzD ), where rigorous mathematical proofs of approaches and lemmas are much less common.

---

> ### Comment · Reviewer_4ZDn · 2024-11-29
> **Further Response to Authors**
>
> I appreciate the authors' detailed explanation, which has clarified the focus and methods of this paper. I do not have any questions regarding the methodology. However, since the methods appear relatively straightforward, I wondered if there are any underlying theoretical foundations for this approach, particularly concerning its effectiveness and efficiency. Incorporating such insights could significantly enhance the impact of this work. I have increased my rating previously; however, I hope my suggestion has highlighted some potential aspects that might be missing in this paper for both the authors and the broader audience.

---

### Author Response · Authors · 2024-11-20
**General Response to All Reviewers**

We appreciate the time the reviewers took to understand and evaluate our work. Here, we present a summary of the major changes in the new version of the manuscript and address selected review questions. We are also posting individual responses to the reviewers’ comments.

→ We added a ‘summary of contributions’ to the **Introduction** since reviewers made us aware that the extent of our contributions was initially unclear:

1. We develop an efficient GNN-based model for electronic property prediction by combining (1) the SO(2) convolution approach detailed in [1], (2) the equivariant attention mechanism proposed in [2], and (3) concepts from [3] and [4] to introduce learnable node/edge embeddings along with a basis transformation layer to pre-process the targets and map predictions to the Hamiltonian output. We provide the code for this implementation [in the Supporting Materials].

2. We propose an efficient augmented partitioning method that breaks down input graphs into small pieces and corrects atomic environments with masked virtual nodes and edges. This allows arbitrarily large graphs to be decomposed into independent partitions that can fit into GPU memory during training without compromising the achievable testing accuracy. Our approach enables the training and prediction of unfeasibly large systems including realistic amorphous materials and heterostructures that can contain up to hundreds of thousands of atoms in a unit cell.

→ We added more details on the near-sightedness principle of the Hamiltonian, including a new **Fig. 5 (b)**, which shows the decay in the magnitude of the matrix elements as a function of the interatomic distance.

→ We added a new **Appendix I** with further material examples as requested by multiple reviewers. Specifically, we added two examples of disorder (Sub-stoichiometric HfOₓ and Pt-doped Amorphous Germanium), to further demonstrate the strength and robustness of our approach, as well as one example of crystalline HfO₂. We note that these types of structures are standard components of our materials research workflows and all training data was developed for downstream applications of this work.

→ Appendix I also provides an explanation of why our prediction accuracy is affected in part by limited training data. In contrast to existing datasets offering Hamiltonians for small molecules and crystalline materials, training data produced for our work is far more computationally expensive to generate (see **Appendix G.3** for a breakdown of the 100x runtime difference to generate electronic Hamiltonians for 3-atom H₂O and 3000-atom HfO₂). We show, however, that using two training structures instead of one decreases the prediction error from **5.87 meV** to **3.55 meV** on the HfO₂ example, demonstrating that the accuracy can be further improved with more data.

**References**

[1] Passaro, S. &; Zitnick, C.L.. (2023). Reducing SO(3) Convolutions to SO(2) for Efficient Equivariant GNNs. *Proceedings of the 40th International Conference on Machine Learning, Proceedings of Machine Learning Research* 202:27420-27438

[2] Yi-Lun Liao, Brandon Wood, Abhishek Das, and Tess Smidt. Equiformerv2: Improved equivariant transformer for scaling to higher-degree representations, 2023.

[3] Gong, X., Li, H., Zou, N. et al. General framework for E(3)-equivariant neural network representation of density functional theory Hamiltonian. Nat Commun 14, 2848 (2023).

[4] Yuxiang Wang, He Li, Zechen Tang, Honggeng Tao, Yanzhen Wang, Zilong Yuan, Zezhou Chen, Wenhui Duan, and Yong Xu. Deeph-2: Enhancing deep-learning electronic structure via an equivariant local-coordinate transformer, 2024a

---

### Meta-Review · Area_Chair_rXnN · 2024-12-22

**Metareview:**

This paper studies the prediction of Hamiltonian matrix using equivariant networks. This problem has been studied for molecules and crystal materials, and this work tries to extend the task to other disordered materials. The paper received weak support from the reviewers. Among the four reviewers, three of them are marginally positive, while the other one is many concerns, some of which have been partially addressed during rebuttals while others remain. After reading the paper and the reviews and discussions, I feel this is a borderline paper that may benefit from a major revision. Specifically, the addressed task is a bit narrow as previous work has studied this problem for molecules and crystals, and this work extends it to disordered materials. Also the method is quite straightforward, and the authors admit that it is a combination of a few prior approaches. One of the reviewers asked for theoretical justifications, which were added during discussions, but not to the satisfaction of reviewer.

**Additional Comments On Reviewer Discussion:**

The theoretical justification provided by the authors during discussions is not convincing, and the reviewer is still concerned after discussions.

---

### Decision · Program_Chairs · 2025-01-22

Reject